# Compact artificial neuron based on anti-ferroelectric transistor

Rongrong Cao [1,4], Xumeng Zhang [2,4], Sen Liu [1,4], Jikai Lu[3], Yongzhou Wang[1], Hao Jiang[2], Yang Yang[3], Yize Sun[3], Wei Wei[3], Jianlu Wang[2], Hui Xu[1], Qingjiang Li [1] ✉ & Qi Liu [2] ✉

Neuromorphic machines are intriguing for building energy-efficient intelligent systems, where spiking neurons are pivotal components. Recently, memristive neurons with promising bio-plausibility have been developed, but with limited reliability, bulky capacitors or additional reset circuits. Here, we propose an anti-ferroelectric field-effect transistor neuron based on the inherent polarization and depolarization of $Hf_{0.2}Zr_{0.8}O_2$ anti-ferroelectric film to meet these challenges. The intrinsic accumulated polarization/spontaneous depolarization of $Hf_{0.2}Zr_{0.8}O_2$ films implements the integration/leaky behavior of neurons, avoiding external capacitors and reset circuits. Moreover, the anti-ferroelectric neuron exhibits low energy consumption (37 fJ/spike), high endurance (>$10^{12}$), high uniformity and high stability. We further construct a two-layer fully ferroelectric spiking neural networks that combines anti-ferroelectric neurons and ferroelectric synapses, achieving 96.8% recognition accuracy on the Modified National Institute of Standards and Technology dataset. This work opens the way to emulate neurons with anti-ferroelectric materials and provides a promising approach to building high-efficient neuromorphic hardware.

In the past few decades, neuromorphic computing, mimicking the human brain's architecture and operation with electronic devices, has attracted great interest due to its high biomimetic and high-energy efficiency[1–3]. Artificial neurons are the core components of neuromorphic computing implementation, emulating biological neurons functions of potential accumulation and firing[4,5]. For the hardware implementation of neurons, hardware overhead, energy efficiency, and reliability are the critical evaluation criteria[5,6]. Yet, current hardware demonstrations of neurons struggle to satisfy these key metrics simultaneously.

Generally, the complementary metal-oxide-semiconductor (CMOS) circuit is the most mature and stable scheme for emulating biological neurons. Nevertheless, due to the lack of intrinsic biological resemblance and the complexity of circuits, CMOS neurons face many

challenges in density or energy efficiency[7–9]. Recently, various emerging devices have been extensively explored to emulate biological neurons benefiting from their biological resemblance and scalability. Memristive neurons have trigged the most interest among them, including redox memristors[10–13], Mott memristors[14–19], phase-change memristors (PCM)[20–22], magnetic random access memory (MRAM)[23,24], etc. These neurons utilize the gradual switching of conductance to mimic membrane potential evolution, successfully emulating essential biological neuron functions with low hardware cost. However, high electroforming voltage and limited reproducibility due to temporal and spatial variations are still open questions[25,26]. In addition, capacitors are usually needed to realize the integration in memristive neurons, which limits their practical applications in large-scale neuromorphic computing systems[10,17,27]. In the very recent research,

[1]College of Electronic Science and Technology, National University of Defense Technology, 410073 Changsha, China. [2]Frontier Institute of Chip and System, Fudan University, No. 2005 Songhu RoadYangpu District 200438 Shanghai, China. [3]Key Laboratory of Microelectronic Devices & Integrated Technology Institute of Microelectronics, Chinese Academy of Sciences, No. 3 Beitucheng West RoadChaoyang District 100029 Beijing, China. [4]These authors contributed equally: Rongrong Cao, Xumeng Zhang, Sen Liu. ✉e-mail: qingjiangli@nudt.edu.cn; qi_liu@fudan.edu.cn

novel ferroelectric polarization-based neurons are proposed and experimentally demonstrated[28–32]. They utilize gradual polarization to mimic the integration process of biological neurons without additional capacitors[33]. Moreover, polarization is the intrinsic property of ferroelectric materials, which is recognized to be reproducible, reliable, and energy-efficient[29,34]. These features are promising to implement neurons. However, ferroelectric devices are nonvolatile, and thus need a feedback path[28–30] or a special design of ferroelectric layer[31,32] to achieve spontaneous reset after firing. The feedback path will increase the hardware cost and energy consumption of neuron implementation. In addition, it will increase the complexity of the operation, as each new input must wait for the completion of the previous reset process, especially in a system with a rate coding scheme. Thus, demonstrating an ideal electronic device that processes advanced and balanced neuronal performance without additional capacitors and reset feedback path deserves more attention.

In this work, we report a leaky integrate-and-fire (LIF) neuron based on a CMOS-compatible anti-ferroelectric field-effect transistor (AFeFET). The intrinsic polarization/depolarization processes of the $Hf_{0.2}Zr_{0.8}O_2$ AFeFET successfully emulate the integrate/leaky neuronal functions without any capacitors and reset peripheral circuits. Furthermore, attributing to the plentiful merits of ferroelectric materials, AFeFET neuron exhibits many superiorities: electroforming-free, ultra-low-energy consumption (37 fJ/spike), high endurance ($>10^{12}$), small cycle-to-cycle variation (as low as 3.93%) and device-to-device variations (7.57%). Also, we present that the temporal integration speed in such an AFeFET neuron depends on the intensity of postsynaptic potential, illustrating the fundamental features for performing classification tasks. Subsequently, we demonstrate a two-layer spiking neural network (SNN) with full-ferroelectric architecture for learning and recognizing the Modified National Institute of Standards and Technology (MNIST) datasets by simulation, obtaining the maximum recognition accuracy 96.8% comparable to ideal neurons. These results demonstrate that the proposed AFeFET neuron is a competitive candidate for constructing neuromorphic systems.

## Results

### Volatile AFeFET as LIF neurons

Figure 1a shows the architecture and processing model of biological neurons. In specific, neuronal dendrites receive input spike information from pre-neurons and transmit it to soma. Then soma integrates information and triggers an action potential when the membrane potential reaches a threshold value. The axon transmits the generated action potentials to post-neurons, and the membrane potential depolarizes to a resting state[5]. The increase/decrease of membrane potential corresponds to the opening or shutting of $Na^+/K^+$ channels, corresponding to three stages in Fig. 1b. Here, the dynamic process of membrane potential can be mimicked vividly by the intrinsic polarization/depolarization of anti-ferroelectric (AFE) materials. Under a silent state, AFE materials have spontaneous polarizations, but the orientations of adjacent diploes are opposite, resulting in zero net macroscopic remanent polarization, as shown in the inset of Fig. 1c. However, the diploes can be aligned by the sufficient electric field, and the phase switches from AFE to ferroelectric (FE)[35]. Usually, the electric field-induced FE phase is not stable, which will recover to AFE phase when the electric field is released. Thus, AFE materials exhibit

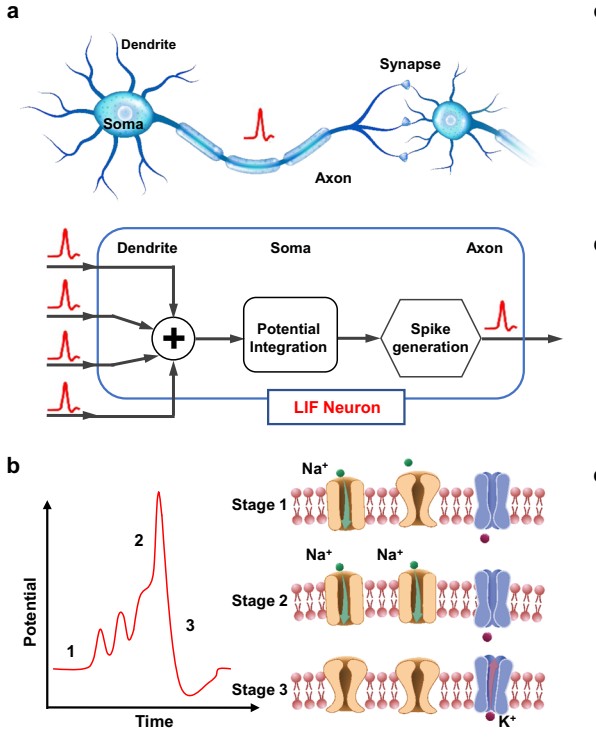

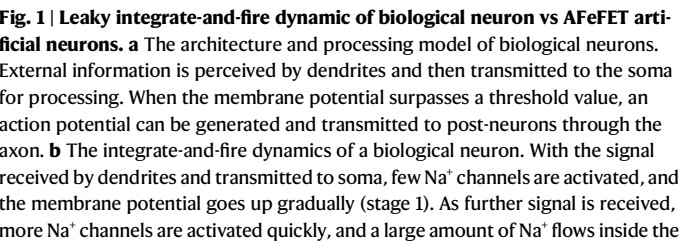

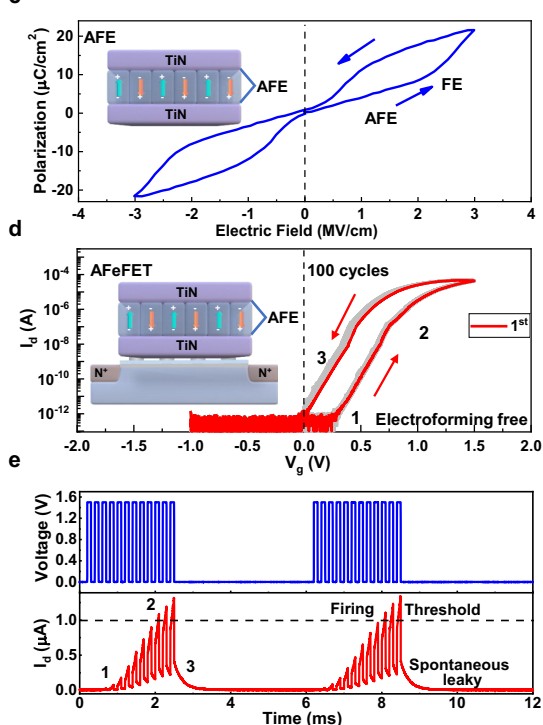

**Fig. 1 | Leaky integrate-and-fire dynamic of biological neuron vs AFeFET artificial neurons. a** The architecture and processing model of biological neurons. External information is perceived by dendrites and then transmitted to the soma for processing. When the membrane potential surpasses a threshold value, an action potential can be generated and transmitted to post-neurons through the axon. **b** The integrate-and-fire dynamics of a biological neuron. With the signal received by dendrites and transmitted to soma, few $Na^+$ channels are activated, and the membrane potential goes up gradually (stage 1). As further signal is received, more $Na^+$ channels are activated quickly, and a large amount of $Na^+$ flows inside the membrane, inducing the membrane potential goes up rapidly (stage 2). Once the membrane potential exceeds a certain threshold, it decreases due to the inactivation of $Na^+$ channels and the opening of $K^+$ channels (stage 3). **c** The representative double hysteresis of AFE materials. The zero net macroscopic remanent polarization indicates the volatile characteristics of AFE materials. **d** The typical transfer curves of AFeFET, exhibiting counterclockwise hysteresis and volatility. The arrows indicate the switching sequences. **e** The continuous firing events of AFeFET neuron. After several milliseconds of free time, the AFeFET neuron can restart the LIF process and fire again.

volatile characteristics and representative double hysteresis ($P_r \approx 0$ at 0 MV/cm) as shown in Fig. 1c. We establish the dynamic relation between the intrinsic volatile characteristic of AFE and integrate/leaky neuronal functions by constructing an AFeFET, which is integrated by an AFE capacitor (TiN/Hf$_{0.2}$Zr$_{0.8}$O$_2$/TiN) and a MOSFET (see "Methods" for the details of fabrication processes), as shown in the inset of Fig. 1d. Figure 1d shows that the typical transfer characteristic of AFeFET exhibits volatility, which differs from that of nonvolatile FeFET at $V_g = 0$ (Supplementary Fig. 1).

The volatility of AFeFET emulates the self-recovery of biological neurons, which helps avoid external peripheral reset circuits. Furthermore, the AFeFET device is electroforming-free, which saves an additional high-voltage forming circuit. In order to investigate the LIF function of AFeFET, continuous gate pulses (100 μs width, 100 μs interval, 1.5 V amplitude) representing postsynaptic potentials are applied to the gate of the AFeFET. The corresponding drain current ($I_d$) representing the membrane potential demonstrates replicable LIF behavior under gate pulse trains (Fig. 1e). The $I_d$ increases under the excitatory spike trains and the neuron fires when $I_d$ reaches a threshold (1 μA). After firing, $I_d$ decreases spontaneously in a millisecond of free time eventually, which means the AFeFET neuron recovers and gets prepared for the next firing. These merits make AFeFET suitable for emulating the integration and recovery process of neurons.

## Device characteristics and mechanism of the AFeFET

In this work, the volatile characteristics of the AFeFET neuron are dominated by the composition of zirconium in Hf$_x$Zr$_{1-x}$O$_2$ film. The Hf$_x$Zr$_{1-x}$O$_2$ exhibits paraelectric-FE-AFE transition with increasing the concentration of zirconium element (0–100%) (Supplementary Fig. 2). Actually, HfO$_2$ and ZrO$_2$ exhibit very similar physical and chemical properties, such as crystal phases, lattice parameters except the dielectric properties. The pure HfO$_2$ shows linear dielectric characteristics under electric field due to the centrosymmetric monoclinic structure[36]. The FE properties occur with the increasing zirconium content in doped HfO$_2$, which is induced by the existence of a noncentrosymmetric o-phase structure. The hafnium-rich ferroelectric Hf$_x$Zr$_{1-x}$O$_2$ oxides exhibit nonvolatility[37–39], thus generally serve as memory materials. With further increasing the zirconium content, the volatile AFE properties occur in zirconium-rich Hf$_x$Zr$_{1-x}$O$_2$ oxides. The polarization of AFE can be triggered by an electric field and increases under a higher electric field. But it can still revert to the initial state as the applied electric field is removed (Supplementary Fig. 3). Usually, the polarization of AFE can be ascribed to the phase transition from AFE to FE phase under the influence of electric field. The electric field-induced phase transition is always accompanied by a large-volume change[40]. When the electric field is released, the induced FE phase will recover to the AFE phase due to the strains resulting from volume expansion[38,41,42]. As a result, the zirconium-rich Hf$_x$Zr$_{1-x}$O$_2$ oxides exhibit intrinsic volatility. This is the charm of AFE materials used for constructing artificial neurons.

Figure 2a, b shows the plane structure and the detailed cross-sectional image of the AFeFET neuron. According to these images, the structure of the AFeFET neuron can be observed clearly, in which an AFE capacitor (yellow square) integrates on the gate of a conventional MOSFET. The energy-dispersive X-ray spectroscopy (EDS) mapping and line scan EDS were performed to further identify the elements and structure of AFeFET (Fig. 2c, d). The thickness of TiN/Hf$_x$Zr$_{1-x}$O$_2$/TiN is 40 nm/10 nm/40 nm, and the interfaces of all layers are clean and flat. In addition, the Hf, Zr, Ti, N, W elements distribute uniform and are free of inter-diffused. Then we focus on investigating the characteristics of AFE layer due to its dominant role in neuronal behavior. The composition of AFE layer is controlled by alternate deposition (one cycle HfO$_2$ and four cycles ZrO$_2$), and is confirmed (hafnium:zirconium ≈1:4) by the peak areas and the relative sensitivity factors in X-ray

photoelectron spectroscopy (XPS) results (Supplementary Fig. 4). The high-resolution transmission electron microscopy (HRTEM) image for the details of the AFE films is presented in Fig. 2e. The polycrystalline nature and the lattice fringes of different crystals can be observed clearly. Figure 2f, g depicts the crystal structure and corresponding fast Fourier transform (FFT) image of the white square area in Fig. 2e. The relative angle and distance between two lattice planes and diffraction spots indicate the existence of [0-10]-oriented AFE tetragonal $P4_2/nmc$ phase. In addition, the arrangement of zirconium atoms (green dots) is very regular, and the relative angle and lattice constants are measured directly as 55.6°, 3.49 Å, and 3.2 Å, respectively. These zirconium atoms parameters match the atomic model of [0-10]-oriented plane of tetragonal $P4_2/nmc$ phase exactly in Supplementary Fig. 5. These results confirm the existence of t-phase in AFE films, which is the foundation of the AFeFET neuron.

In order to demonstrate the dynamic of AFeFET neuron, the mechanism is shown in Fig. 2h, which is related to the transformation between AFE and FE domains. At stage 1, several input pulses as postsynaptic signals are applied to the gate of AFeFET neuron, and the electric field-induced FE orthorhombic phase (o-phase) domains nucleate, which transform from AFE tetragonal phase (t-phase) domains under the gate pulse stimuli. The polarized charges accumulate in the AFE layer and modulate the channel resistance of MOSFET, resulting in the $I_d$ begins stepping up gradually. This process is just as the small portion of Na$^+$ channels opening. With further applying gate pulses, it comes to stage 2, at which the electric field-induced FE o-phase domains grow and expand. As a result, more attracted electrons accumulate in the channel of AFeFET, and the $I_d$ increases greatly, corresponding more Na$^+$ channels opening. Once the $I_d$ surpasses the threshold, the AFeFET neuron would fire. After firing, the electric field-induced FE o-phase domains transform back to AFE t-phase domains due to the release of gate pulse. Consequently, the attracted electrons discharge and the channel of AFeFET switches off, indicating the AFeFET neuron returns to resting potential (stage 3). This process corresponds to the opening of K$^+$ channels in biological neurons. Then, the AFeFET neuron fires again under another gate pulse stimuli. This repeatable and stable electric field-induced phase transition accounts for the intrinsic neuronal resemblance of AFeFET. The atomic-scale phase transition between t-phase and o-phase under the influence of electric field has been observed clearly by Lombardo et al. via in situ HRTEM[42].

To present the gradual electric field-induced phase transition of AFE t-phase, we investigate the tendencies of $I_d$ under continuous gate pulses. Figure 2i shows the tendencies of $I_d$ under different gate pulse amplitude, while the gate pulse interval and width are fixed to 100 μs, respectively. Under the first 20 continuous gate pulse stimuli, an obvious integration process of $I_d$ can be observed. This resulted from the gradual formation of electric field-induced o-phase, which induces the electrons accumulation in the channel of AFeFET. The gate pulses with larger amplitude result in more o-phase e domains formation and quicker growth of $I_d$. With further gate pulse stimuli, the reversible domains tend to reach a saturation regime. This represents the dynamic balance between the electric field-induced phase transition and the recovery of the AFE t-phase, and the $I_d$ does not increase anymore. Compared between stimuli with different amplitudes, it is clearly that the higher pulse amplitude will lead to faster growth speed and a larger saturation value of $I_d$, which is because that more AFE t-phase domains can be switched to FE o-phase domains. Noting that a similar tendency of $I_d$ can be observed under different gate pulse intervals and widths, as illustrated in Supplementary Fig. 6. Input stimuli pulses with shorter pulse intervals or wider widths induce faster integration speed and larger saturation value of $I_d$. In all cases, the $I_d$ increases gradually and then tends to saturate corresponds to the gradual electric field-induced phase transition and saturation processes of AFE t-phase. In addition, the tendencies of $I_d$ also illustrate the

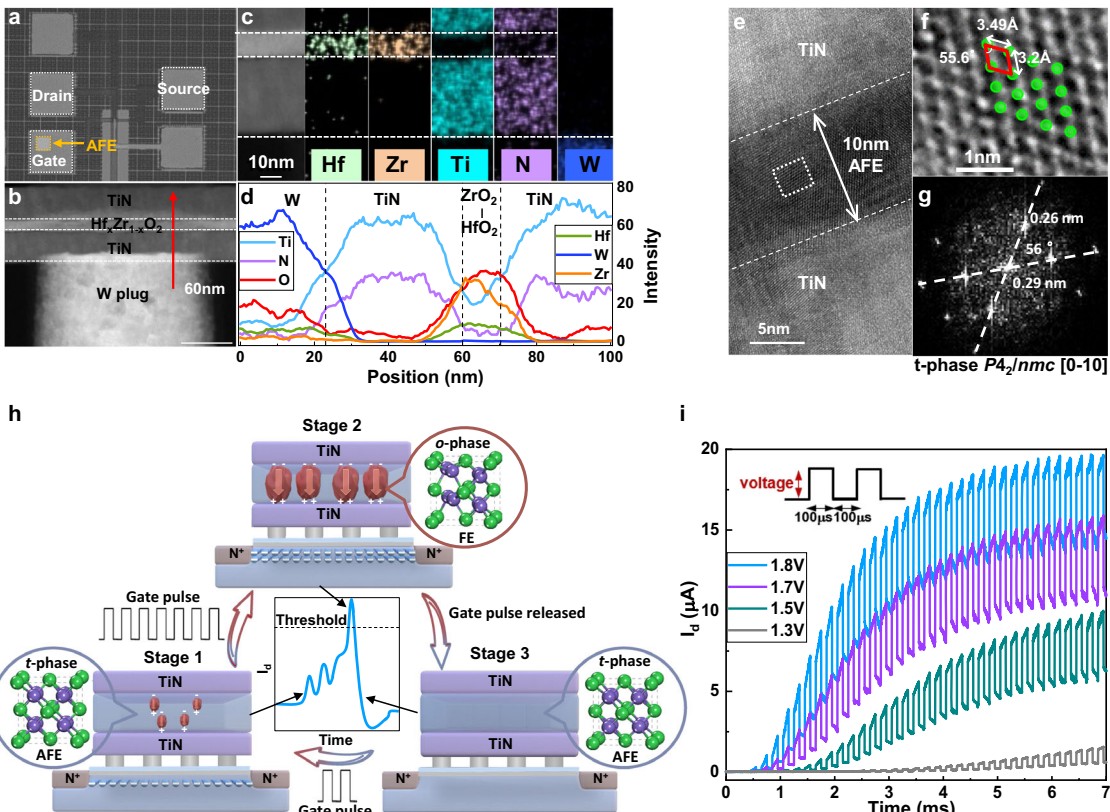

**Fig. 2 | Device characteristics and mechanism of AFeFET neurons. a** Planer structure of AFeFET device. **b** The cross-sectional image of AFeFET device. **c** The elemental mapping of the materials in the system for Hf, Zr, Ti, N, and W, respectively. **d** The line scan EDS of the device cross-section, which corresponds to the red arrow in (**b**). **e** The HRTEM image of the AFE films. **f** The crystal structure of the domain, which corresponds to the white square area in (**e**). **g** The FFT image of the crystalline domain in (**f**). The measured angles and distances between relative crystal faces, and diffraction spots suggest the existence of [0-10] oriented tetragonal $P4_2/nmc$ structure. **h** The integrate-and-fire dynamics of AFeFET neuron. At stage 1, the electric field-induced FE domain nucleates, and the electrons begin to accumulate in the channel of AFeFET. With the further gate pulse applied (stage 2), the electric field-induced FE domains grow and expand, resulting in more attracted electrons accumulate in the channel of AFeFET. At stage 3, the electric field-induced FE domains transform back into the AFE domains when the gate pulse electric field is removed. The channel AFeFET switches off, and the neuron backs to resting potential. **i** The tendencies of $I_d$ under continuous gate pulses with different amplitudes (1.3–1.8 V amplitude, fixed 100-µs interval, fixed 100 µs width).

intrinsic plasticity of neurons[43], demonstrating that the AFeFET has high potentiality for hardware implementation of artificial LIF neurons.

## Neuronal characteristics of the AFeFET

To investigate the strength-modulated integration process of AFeFET neurons, we apply gate pulse stimuli with different amplitudes and widths to implement LIF neuron functions, as shown in Fig. 3a, b. As the input pulse intensity (amplitude/width) increasing, the AFeFET neuron needs fewer input spikes to reach the threshold (1 µA), which indicates a higher firing rate under stronger stimuli strength. This is because more electric field-induced o-phases are formed under stronger pulse intensity, resulting in faster charge integration speed. Correspondingly, a higher $I_d$ of the AFeFET neuron needs longer time to leak (Supplementary Fig. 7), which could be clarified as adaptive recovery. To further study how the leaky behavior influences the integration process, we measure the $I_d$ under gate pulse stimuli with different intervals (50–600 µs), as shown in Fig. 3c. As the interval increasing, more input pulses are needed to integrate the $I_d$ to reach the threshold value. This is because that more charges are released during the free interval time, and more input pulses are required to compensate for that. It is worth noting that, when the interval time is wider enough, the $I_d$ cannot reach the threshold anyway. This feature represents the filtering capability of the neuron for weaker input signals, which is important in biological systems and neuromorphic systems[44]. To further evaluate the stability of the AFeFET neuron, we extract the statistical data of input spike

numbers for firing as a function of input amplitudes, as shown in Fig. 3d. The firing event needs fewer input spikes and tends to be more stable as the stimuli intensity increasing. This phenomenon exhibits that the AFeFET neuron performs high-precision computation under enough stimuli intensity, which is favorable for performing high-precision tasks. A similar relationship between input spike numbers for firing and pulse widths (or intervals) is observed, as shown in Fig. 3e, f, respectively. To directly present the stability of the AFeFET neuron, we calculate the standard deviation ($\sigma$) of integration pulse number under each stimuli condition and label them out in Fig. 3d–f. Furthermore, the cycle-to-cycle variation is calculated by dividing standard deviation ($\sigma$) by mean value ($\mu$). The lowest variation (3.93%) between cycles is obtained under 1.7 V gate pulse amplitude, 50-µs interval, and 100 µs width. It should be noted that optimizing the gate pulse parameters may further enhance the uniformity between cycles. These results demonstrate that the AFeFET neurons can successfully emulate the strength-modulated spike frequency characteristics of biological neurons with high stability, making the AFeFET neurons firstcapable of carrying out the classification tasks[11,45].

To further access the compatibility of the AFeFET neuron for implementing unsupervised learning, we investigated the lateral inhibitory property. During the accumulation of AFeFET neuron membrane potential, the excitability will be inhibited immediately when the AFeFET neuron receives inhibitory stimuli from adjacent neurons, as shown in Supplementary Fig. 8a, b. Moreover, the AFeFET neuron

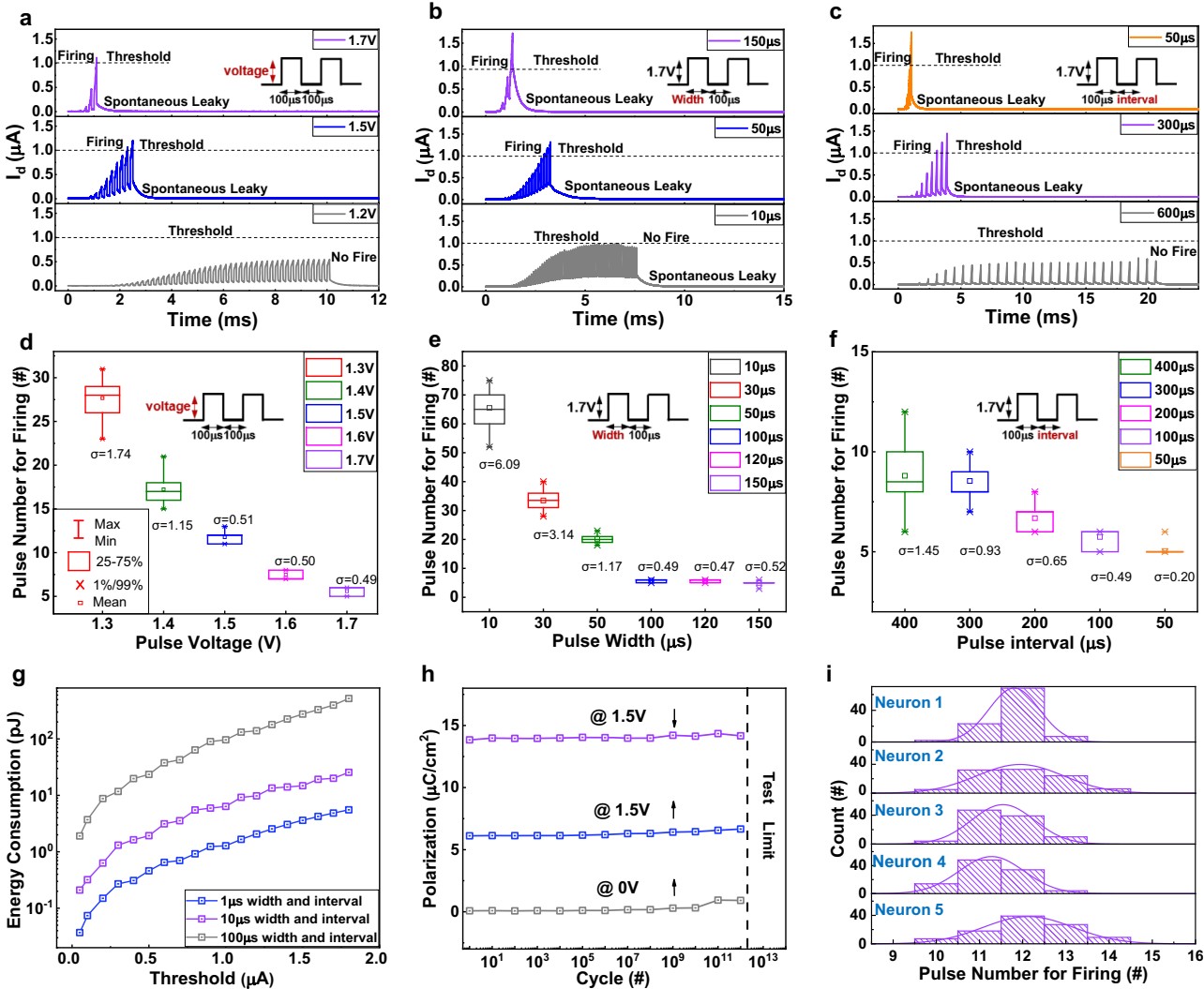

**Fig. 3 | AFeFET LIF neuron device characteristics.** The dynamic of the strength-modulated integration process of AFeFET neuron under (**a**) different gate pulse amplitudes (1.2–1.7 V amplitude, fixed 100 µs interval, fixed 100 µs width), **b** different gate pulse widths (fixed 1.7 V amplitude, fixed 100 µs interval, 10 µs–150 µs width), **c** different gate pulse intervals (fixed 1.7 V amplitude, 50–600 µs interval, fixed 100 µs width). When the input pulse intensity is not enough, the AFeFET neuron cannot fire. Statistical data of input pulse numbers of AFeFET neuron firing as a function of **d** amplitudes, **e** widths, **f** intervals. The firing event needs fewer input spikes and tends to be more stable as the stimuli intensity increasing. Each data point is collected from 50 firing activities. **g** The energy

consumption dependence on pulse parameters and threshold values of AFeFET neurons. The energy consumption decreases remarkably as the threshold and pulse-width (interval) decrease. The experimental lowest energy consumption is 37 fJ per spike and can be further reduced as decreasing pulse parameters (threshold). **h** The AFeFET neuron can fire more than $10^{12}$ cycles stably without significant deterioration. The results are extracted from polarization-voltage (PV) hysteresis loops @0 V forward field, 1.5 V forward field, 1.5 V backward field, respectively. **i** Statistical data of input pulse numbers for AFeFET neuron firing. Data are collected from 100 firing activities for each AFeFET neuron.

needs more excitatory inputs for the next firing under stronger lateral inhibition intensity (Supplementary Fig. 8c). This behavior is similar to the suppressive phenomenon in biological neurons between each other, which is valuable for performing competitive learning tasks.

For artificial neurons, low-energy consumption, high endurance, and high reliability are critical merits. To investigate the energy consumption of the AFeFET neuron, we performed a systematic analysis under different input pulse parameters and threshold values, as shown in Fig. 3g (extracted from Supplementary Fig. 9). The lowest energy consumption of 37 fJ/per spike can be obtained under the 50 nA threshold, 1 µs pulse width, and 1-µs pulse interval. Furthermore, the energy consumption decreases remarkably as the threshold and pulse-width (interval) decrease. Thus, it is reasonable to infer that the energy consumption can be further reduced by decreasing the threshold and pulse width (interval). Moreover, the AFeFET neurons demonstrate considerable repeatability. Supplementary Fig. 10a shows $5 \times 10^5$ stable

firing cycles of the AFeFET neuron without any significant deterioration. In order to speed up the measurement, the AFE MIM structure, which is the endurance bottleneck of AFeFET, is measured for higher endurance ($10^{12}$ cycles) (Supplementary Fig. 10b). Based on the endurance measurements above, it is reasonable to believe the AFeFET neuron could support more than $10^{12}$ firing events (Fig. 3h). Figure 3i shows the histograms of input spike numbers for firing, which are collected from 100 firing activities of each AFeFET neuron. The required pulse number for firing is concentrated nearby 12, and the device-to-device variation is calculated to be as low as 7.57%. This variation is extracted from 500 firing activities in five AFeFET neurons. As we claimed before, the uniformity could be further enhanced by optimizing the gate pulse parameters. These results indicate that the AFeFET neuron has high uniformity and great potential in large-scale applications.

The basic integration and fire functionality of the proposed neuron can be achieved by only one AFeFET, while the examination of

**Table 1 | Comparison of various hardware implementations of spiking neurons**

| Mechanism | Variation[#] | Structure | Endurance | Energy/spike | Hardware | Self-reset | Driven capability |
|---|---|---|---|---|---|---|---|
| Redox | 9.63%–31.4%[10,48–51] | SiO$_x$N$_y$:Ag[10] | >10$^6$ | >60 pJ* | 1 C + 1 T + 1ED | √ | x |
| | | Ag/SiO$_2$[11] | >10$^8$ | ~500 nJ* | 1 C + 2 T + 1ED | √ | x |
| | | Ag/HfO$_2$[13] | —— | 18 pJ | 1 C + 1 T + 1ED | √ | x |
| Mott | 6.31%–11.9%[17,52–54] | NbO$_2$[13] | —— | ~52 pJ | 1 C + 1 T + 1ED | √ | x |
| | | NbO$_x$[14] | >10$^{12}$ | ~3 pJ* | 1 T + 1ED | √ | x |
| | | GaTa$_4$Se$_8$[18] | —— | ~10 uJ* | 1 T + 1ED | √ | x |
| Phase-change | 6.82%–12.5%[20,55] | GST[20] | 3 × 10$^9$ | >50 pJ | 19 T + 1ED | x | √ |
| | | GST[22] | —— | 10 pJ* | >1 C + 7 T + 1ED | x | √ |
| Magnetic | 3%–10%[56–58] | MTJ[23] | —— | ~7 fJ (simulation) | >4 T + 2ED | x | √ |
| | | STT-MRAM[24] | —— | —— | >21 T + 1ED | x | √ |
| Ferroelectric | 1%– 4.03%[32,59,60] | FeFET[30] | —— | ~360 pJ (simulation) | 3 C + 9 T + 1ED | x | √ |
| | | FeFET[29] | —— | 1–10 pJ | 1 C + 6 T + 1ED | x | √ |
| | | Leaky-FeFET[32] | —— | ~420 pJ* | 2 T + 1ED | √ | x |
| Anti-ferroelectric | <3.93%** | AFeFET | >10$^{12}$ | 37 fJ | 8 T + 1ED | √ | √ |

[#]The variation results are rarely reported in literature. In this table, the variation data are obtained from devices with the same mechanism category.

*The energy consumption per spike is calculated approximately from the $I$–$t$ and $V$–$t$ curves in these reference papers, respectively.

**With further optimizing the stimuli pulse parameters.

√ has this property.

x has no such property.

To unify the benchmark of hardware overhead, all the circuit components in these reference papers are equivalent to three categories: capacitor (C), emerging device (ED) and transistor (T). In chip manufacturing, the area of a resistor is equivalent to that of a transistor. 1 latch is composed of four transistors. 1 XOR gate is composed of ten transistors. An integrated operational amplifier usually consists of more than 20 transistors.

current threshold, the generation of output spike, and controllable refractory period need additional circuits as shown in Supplementary Fig. 11. The detail of this circuit design is described in the supporting information. For clearly presenting the comprehensive merits of our AFeFET neuron, a benchmark comparison with other typical neurons based on emerging devices is summarized in Table 1. Considering the energy consumption, endurance, and hardware overhead, which are the critical evaluation criteria of the artificial neurons, the AFeFET neuron exhibits attractive performances.

## Network-level performance of AFeFET neuron

We have demonstrated that the AFeFET neurons can provide better energy efficiency and higher uniformity compared to the other neurons based on emerging devices. It is also essential to evaluate the network-level performance using the AFeFET neuron for the hardware implementation of SNNs. Subsequently, we construct a two-layer SNN (784 × 400 × 10) for classifying MNIST datasets, as shown in Fig. 4a. In this network, we adopt a time-to-first-spike coding method, in which all input neurons fire exactly one spike per stimulus, but the firing order carries information. A larger input corresponds to an earlier spike of the neuron. In the output layer, the first fired neuron determines the class of stimulus. As soon as one of the output neurons fires, the network assigns the corresponding category to the input, and the inference process stop. Thus, such a coding scheme is much more suitable for hardware implementation. The right panel of Fig. 4a presents the proposed hardware implementation of the network based on ferroelectric field-effect transistor (FeFET) synapses and AFeFET neurons. During inference, the input signal is applied to the drains of FeFET synapses (BLs), pulse-width modulators (PWMs) collect current on source lines (SLs) and convert to pulses with fixed amplitude and various widths. The outputs of PWMs serve as postsynaptic potentials and be applied to the gates of AFeFET neurons for performing the integration process. Then, we train such a network to learn MNIST datasets for illustrating the feasibility. During training, we adopt a supervised temporal backpropagation algorithm proposed by Kheradpiseh[46]. Figure 4b shows the training results with the AFeFET neurons under 1 μA threshold, achieving ~95% recognition accuracy.

These results demonstrate that our AFeFET neurons have great potential to be used for fabricating SNNs chips.

During training, the threshold value determines the number of integrated inputs (the number of membrane states) and thus affects the network performance. Then, the relation between the threshold value and network performance is further investigated as shown in Fig. 4c. It should be noted that the recognition accuracy is related to the threshold, with the highest 96.8% accuracy under the 2 μA threshold, which is nearly identical to ideal IF neurons. The pulse number for firing is equivalent to the number of the membrane potential during training. When increasing the threshold, the number of potential membranes increases, corresponding to the increasing precision of the membrane potential. Thus, with increasing the threshold, the recognition accuracy increases. Nonetheless, the inference time increases with increasing the threshold value because more integration number is required to trigger neuron firing at higher threshold cases. Thus, there should be a trade-off between recognition accuracy and inference time, and an appropriate threshold value should be selected to balance the network performance in practical applications. Fortunately, the network can still achieve high accuracy (>86%) even at a threshold value low to 62.5 nA, which is favorable for applications that need faster inference time but not rigorous accuracy. As we claimed before, the threshold also affects the energy consumption of the neuron, which is the key parameter for SNNs chip applications. We extract the spike number and energy consumption of the neurons in the system with different training thresholds to evaluate this feature, as shown in Fig. 4d. When the threshold is higher than 0.125 μA, the spike number in the hidden layer (2nd layer) decreases with increasing the threshold. This is because the hidden neurons with a higher threshold are hard to fire. On the contrary, we observe that the total energy consumption of neurons decreases as the threshold decreases. When the threshold reduces to 62.5 nA, the spike number abruptly decreases to be less than 100. This is because under a low threshold value, the winner neuron in the output layer fires earlier, and the network could finish the inference process faster. In that case, only a minority of neurons in the hidden layers fire, and thus the total neurons consume less energy. These results demonstrate that

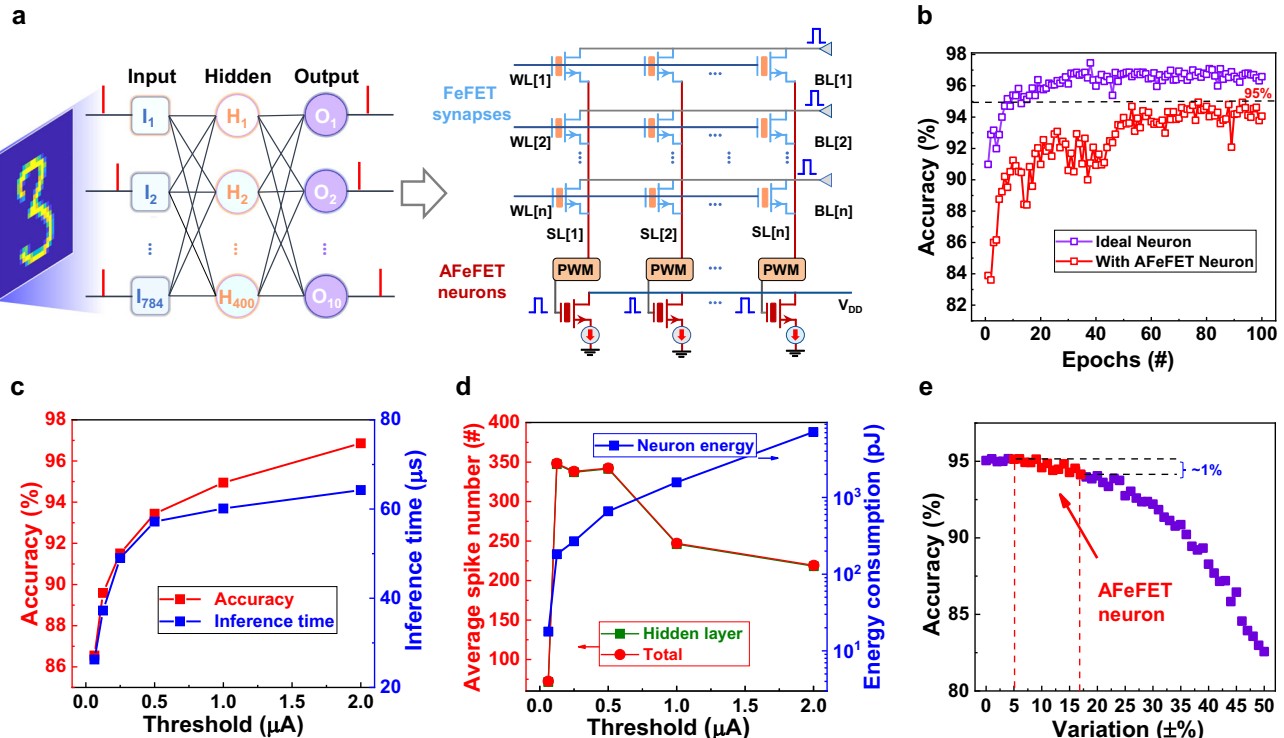

**Fig. 4 | Network-level performance of AFeFET neuron. a** Schematic of two-layer fully ferroelectric SNN (784 × 400 × 10) for classifying MNIST datasets and the proposed hardware implementation of the network based on FeFET synapses and AFeFET neurons. **b** The inference accuracy as a function of training epochs with ideal LIF neurons and AFeFET neurons, 95% recognition accuracy can be achieved for AFeFET neurons under the 1 μA threshold. **c** The speed-accuracy trade-off. The network is trained under different thresholds for all hidden and output neurons, the recognition accuracy and inference time increases with the threshold increasing. **d** The average spike number and energy consumption of the network under different threshold values. The spike number in the hidden layer increases with decreasing the threshold while the energy consumption on neurons decreases (threshold >62.5 nA). This indicates the energy-friendly feature under a lower threshold. **e** The robustness of the network. The network could maintain a high accuracy of ~94.1% even the device variation increases to ±16.7%.

decreasing the threshold supports fast inference speed and could greatly decrease the energy consumption of hardware neurons. In addition, device variation is another important parameter in practical applications. Figure 4e presents the inference accuracy under various device variations after training under the 1 μA threshold. As the red dots are shown, the inference accuracy only decreases 1% even the variation increases to ±16.7%, illustrating nearly no network performance degradation. These results further demonstrate that the proposed AFeFET neuron is suitable for performing SNNs tasks and has great potential for the hardware implementation of SNNs chips.

## Discussion

SNNs, inspired by the human brain, are powerful platforms for enabling low-power event-driven neuromorphic hardware. In SNNs, spiking neurons are the key units that enable spikes, which exchange information through connected plastic synapses. With rich physical dynamics, memristive devices are considered promising devices to emulate spiking neurons. However, the high-energy consumption or limited reliability hinders the applications of memristive neurons in neuromorphic computing.

In this work, we demonstrated a leaky integrate-and-fire neuron based on an AFeFET. The dynamic relationship between the intrinsic polarization/depolarization process of the $Hf_{0.2}Zr_{0.8}O_2$ AFeFET and integrate/leaky neuronal functions are successfully built. The AFeFET neuron features CMOS-compatible, tunable firing frequency, ultra-low hardware cost (no capacitance and additional reset circuit), ultra-low-energy consumption (37 fJ/spike), high endurance (>$10^{12}$), and high uniformity among different cycles and devices, showing advanced overall performances compared with emerging devices-based neurons in literature. To verify the feasibility of the neuron, we constructed a two-layer SNN combined with FeFET synapses, achieving high recognition accuracy (96.8%), low-energy consumption, and high robustness on MNIST datasets. These results demonstrate that the AFeFET neuron is a promising candidate for constructing high-efficient SNN systems and may promote the industrial landing of neuromorphic machines based on anti-ferroelectric materials.

## Methods

### Sample fabrication

The fabrication processes of AFeFET neuron devices are as follows: (1) After ultraviolet lithography and lift-off process, the bottom electrode TiN (40 nm) was deposited on the gate terminal of the NMOS transistor by ion beam sputtering. The NMOS transistor was fabricated by 0.18 μm CMOS technology. The W/L of NMOS is 10 μm/1 μm and its dielectric thickness is 4 nm. (2) Then, 10 nm $Hf_{0.2}Zr_{0.8}O_2$ AFE thin films were deposited on 40 nm-thick TiN bottom electrode by atomic layer deposition (ALD) process at 280 °C substrate temperature. The $Hf[N(C_2H_5)CH_3]_4$, $Zr[N(C_2H_5)CH_3]_4$, and $H_2O$ were used as hafnium precursor, Zr precursor and oxygen source, respectively. The hafnium/zirconium ratio was controlled by alternate deposition of one cycle $HfO_2$ and four cycles $ZrO_2$. (3) Then after the ultraviolet lithography process, 40 nm-thick TiN was grown by an ion beam top electrode was released. The two-terminal metal-insulator–metal structure was integrated on the gate of the NMOS transistor. (4) The fabricated device was annealed for 30 s at 500 °C in nitrogen atmosphere to crystallize.

### Measurement method

The element ratio is confirmed by X-ray photoelectron spectroscopy (ESCALAB 250Xi). The cross-section TEM, high-resolution TEM, energy-dispersive spectroscopy and crystal structure were analyzed by

transmission electron microscopy (FEI Tecnai TF-20, UK). The DC mode is measured by Agilent B1500 semiconductor parameter analyzer. A B1530A fast measurement unit module was used for generating the voltage pulse and measure the response current at the same time. Capacitance-electric field (C-E) tests are performed with 10 kHz AC probing frequency and 30 mV amplitude by Agilent B1500.

## SNN simulation method

In this work, we use time-to-first-spike time coding to encode the input image into a sparse spike train. Each pixel of the input image is encoded into a single spike whose spiking time is inversely proportional to its pixel value. The dense input corresponds to earlier spiking time. And each input pixel will only generate one spike, resulting in sparser spike train than the rate coding method and significantly reducing energy consumption in hardware implementation. We constructed a $784 \times 400 \times 10$ fully ferroelectric SNN for MNIST recognition based on such a coding method. FeFET synapses were considered during training, whose conductance was between 5 μS and 60 μS according to the experimental data in ref. 47. The pulse number (64) for firing under 10 μs pulse width is used due to the highest number counts that correspond to the bits number (6 bits) of the neuron's membrane potentials. The 9% cycle-to-cycle variation is extracted from the statistical data of 10 μs pulse width in Fig. 3e. In addition, the leaky time constant (800 μs) is extracted from the integrate-and-fire process under 10 μs pulse width (Supplementary Fig. 7b). The energy consumption was calculated according to the data of 10 μs pulse width in Fig. 3g. When the neuron reaches the firing threshold, it will emit a spike to the subsequent layer. After emitting a spike, neurons will remain resting state until the end of the time window. In the output layer, the first spiking neuron determines the network decision. Before training, the synapse weights are initialized randomly. In the training process, the target firing time of the correct output neuron is the earliest time that all neurons fire, and the target firing time of other neurons is set to be later than the earliest firing time. According to the defined error function, synapses of the fired neuron before the actual firing time will be updated. In order to update the weights of the hidden layer, a backpropagation algorithm is used to calculate the error of the hidden layer[46]. During training, all time steps need to calculate the error function. In the inference process, the recognition result is obtained when the first spike is generated in the output layer, so there is no need to perform the later time step. Fewer time steps mean lower recognition latency and less energy consumption. By adjusting the neuron's threshold, the spike generation time can be adjusted, resulting in the adjustable recognition speed and energy consumption with acceptable accuracy loss.

## Data availability

All data needed to evaluate the conclusions in the paper are present in the paper and/or the Supplementary Materials. Additional data related to this paper can be requested from the authors. Source data are provided with this paper.

## Code availability

The code of SNN simulation will be available from the corresponding authors upon request.

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

## Acknowledgements

This work was supported by the National Key R&D Program of China under Grant No. 2018YFA0701500, 2019YFB2205102, the National Natural Science Foundation of China under Grant Nos. 62004220, 62004219, 61974164, 62074166, 61804181, 61825404, 61888102, 62104044, 61732020, 61821091, 62104256, and 61851402, the Strategic Priority Research Program of the Chinese Academy of Sciences under Grant XDB44000000 and the China Postdoctoral Science Foundation under Grant No. 2020M673696. The authors thank the valuable discussion from W. Wang, X. Li with National University of Defense Technology, Y. Li and Z.H. Wu with University of Chinese Academy of Sciences.

## Author contributions

R.R.C. and X.M.Z. designed the experiments. R.R.C., X.M.Z., and Y.Z.S. designed and fabricated the AFeFET devices. R.R.C. carried out the electrical experiments. X.M.Z., J.K.L., and Y.Z.W. conducted the simulation. R.R.C., Y.Y., and W.W. carried out the TEM and EDS tests. S.L., H.J., H.X., J.L.W., and Q.L. assisted with data analysis and interpretation. R.R.C., X.M.Z., and S.L. co-wrote the manuscript. All authors discussed the results and revised the manuscript. Q.J.L. and Q.L. supervised the research.

## Competing interests

The authors declare no competing interests.
