## [Peer Review File · Nature Communications]

Compact artificial neuron based on anti-ferroelectric transistorREVIEWER COMMENTS

Reviewer #1 (Remarks to the Author):

The authors claim to demonstrate for the first time an anti-ferroelectric FET (AFeFET) based neuron. In general the topic is interesting and timely, but in my eyes the manuscript lacks significant novelty.

FeFET based neurons have been proposed already by [44] and others. In these works the main ingredient - accumulative switching - has been already demonstrated. However, in the current manuscript under review this aspect stays somewhat vague. Fig. 1c shows no remanent polarization at 0V. However, Fig. 1d shows a remaining MW during sweep. What causes that threshold voltage change - if not the polarization? Also in Fig. 1d it is shown that 1st and 100th cycle result in the very same IDVG. But in Fig. 1e subsequent pulses show increasing Id. That data doesn't fit well together. I would encourage authors to add a plot like fig. 1d but with sweeps just from 0V to 1.5V keeping out the neg. voltages to demonstrate the integration behavior.

The second most important ingredient is the additional functionality of 'self-reset' after firing, which however is not sufficiently demonstrated in the manuscript. Authors don't explain how the neuron would generate a single spike at the output. Moreover, typically the recovery period means a time span where the neuron cannot be excited anymore. But from the data shown it is not obvious how such refractory period can be attained by using only this single device. It seems that authors just stop applying further spikes to the gate after the threshold is reached. It is not shown, how the AFeFET behaves under continuous applied pulses even after reaching a current threshold and sending one spike. Authors should explain what additional circuitry would be mandatory to examine the current threshold, to generate the pulse and to erase the AFeFET.

The system evaluation seems to be based on model assumptions that are not backed by the experiment (see my comments above).

In summary, I cannot recommend the publication of this work in Nature Communications.

Reviewer #2 (Remarks to the Author):

This paper shows a leaky integrate and fire neuron based on antiferro FeFET. Volatility of the device (due to spontaneous depolarization) and spontaneous RESET allows to reproduce neuron leaky behavior, then avoiding the integration of extra capacitances or RESET peripheral circuits. The device is used in a 2 layer SNN for MNIST applications. Low consumption and good endurance is demonstrated.

The paper is complete and presents various topics: material characterization, device electrical characterization and circuit simulation. The neuron emulation is highlighted and well presented. The paper is very clearly written, neuron behavior is clearly described what is helpful for device scientists who may not be expert in the field of brain inspired circuits. The paper is well documented and supplementary materials provide good complementary data. This work would be beneficial for the community as it proposes a device (AFFeFET) showing very promising features (volatility, low endurance...) for LIF neuron emulation.

Some points would need clarification before publication. The paper could thus be published provided that major revisions are provided.

Summary

In the abstract, the authors claim 37fJ/spike, what corresponds to 50nA threshold (fig.3.g). Then the authors claim 96.8% of recognition rate on MNIST, but this corresponds to I_{th} of 2μA (according to fig.4). I think it is misleading as these two features do not correspond to the same operating conditions.

Volatility

Zr content controls the ferro behavior of HfZrO films, in particular this allows to achieve FE or AFE behaviors. Would it be possible to control the retention time, and thus the neuron characteristics (in particular the speed of leaky behavior) playing with the % of Zr?

RESET

How is the neuron RESET after firing? What is the corresponding required overhead in the circuit?

Circuit size

AFFeFET are three terminal devices. They are thus bigger than other emerging devices using 2 terminals. I do not think it is a big issue as number of neurons is less important than number of synapses. However the authors also use a 3 terminal (FeFET) for synapse emulation. Could the authors comment on the impact it can have on the total circuit size?

Current saturation

In page 8, the authors propose interpretation to the saturation of I_d during gate stimuli. I am confused by the interpretation. Indeed, the $I_d(t)$ characteristics looks like typical $I_d(V_g)$ MOSFET curve. As pulses are applied, electric field in the Fe layer increases, acting as a control gate. Thus, $I_d(\text{time})$ curve of fig.2.i could be transferred as a $I_d(V_g)$ curve. Thus, the saturation could be explained by the $I_d(V_g)$ saturation behavior that is measured at high V_g , and I do not see how it is possible to decorrelate the saturation from the transistor channel conduction to the saturation of domain switching. One option may be to test a reference device with same equivalent oxide thickness for the gate oxide but having non ferro behavior. I think it is important to clarify this point to distinguish between current vs ferroelectric saturation.

Current threshold for firing

Various current threshold values are presented in the article. Fig.1 uses $1\mu\text{A}$ and fig.4 shows the threshold values affects recognition rate. Looking at Fig.2, it seems that $I_{th} \sim \mu\text{A}$ is in the sub-threshold regime of the $I_d(V_g)$ characteristics of the transistor. I was this wondering if detecting the threshold in the over threshold regime of the transistor could help in reducing potential variability issue on the number of required pulses to reach a certain current level. As a consequence, I think it would be interesting to quantify the impact of I_{th} value (in sub vs over threshold regime) on the device to device variability of the firing event (ie variability of pulse number for firing vs I_{th} value).

Network hardware implementation (page 12)

In the hardware implementation of the network, during inference, input signal is applied to the gates of the synapses. What signal is applied to the BL? How different would it be to send input signals to the BL?

Recognition rate

Networks aiming at character recognition from MNIST data sets are known to be quite resilient to device variability. Could the authors comment on what would happen if the number of neurons in the hidden layer is decreased, would the system be more sensitive to device imperfections?

Reference

Page 3, line 3: references should be added on structures combining emerging memories with capacitors for neuron implementation.

Gabriel Molas

Reviewer #3 (Remarks to the Author):

In this paper, authors demonstrated a leaky integrate-and-fire (LIF) neuron based on an anti-ferroelectric transistor. The device-level performance was systematically demonstrated. In addition, authors simulated the SNN to demonstrate the feasibility of anti-ferroelectric transistor neurons. However, in this manuscript, the issues of anti-ferroelectric materials for devices were not discussed. In my view, this work is interesting but can be further improved through these comments.

1. In this paper, the phase transition of Hf_{0.2}Zr_{0.8}O_x layer by electric field was used to emulate the LIF characteristics. Hf_{0.2}Zr_{0.8}O_x layer seems to have the potential as neuron devices because of the similarity between intrinsic polarization/depolarization of Hf_{0.2}Zr_{0.8}O_x layer and the dynamic process of membrane in biological neuron. However, Hf_{0.2}Zr_{0.8}O_x layer can show the permanent phase transition from t-phase to o-phase during repeated operation (Adv. Electron. Mater. 6, 2000631, (2020), ACS Appl. Mater. Interfaces 8, 15466-15475 (2016)). This issue should be discussed.
2. As mentioned in comment 1, the permanent phase transition from t-phase to o-phase in Hf_{0.2}Zr_{0.8}O_x layer can be induced during repeated operation. This phenomenon can cause the cycle-to-cycle variation issue. Authors claimed that anti-ferroelectric layer did not show the change during 10¹² cycles in figure 3h. Is there any reason for the absence of the phase transition from t-phase to o-phase during repeated operation? In addition, anti-ferroelectric neurons showed the degradation of output current level after only 5×10⁵ cycles. It seems that anti-ferroelectric neurons cannot be fired afterwards.
3. Authors used NMOS transistor and anti-ferroelectric capacitor to implement anti-ferroelectric transistor. Anti-ferroelectric capacitor was formed on the gate pad of NMOS transistor. In this case, the applied voltage can be divided into the gate dielectric layer of NMOS transistor and anti-ferroelectric layer depending on the capacitance of layers. However, the information of NMOS transistor was missing. The information of NMOS transistor, such as thickness of dielectric and area of gate stack, should be added. In addition, in Figures 1d and 2h, schematic diagrams of device structure were shown as if anti-ferroelectric capacitor was directly stacked on NMOS transistor.
4. As mentioned in comment 3, the applied gate voltage will be divided into the gate dielectric layer of NMOS and anti-ferroelectric layer. Therefore, the applied voltage to anti-ferroelectric layer seems to be lower than 1.5 V, which is average voltage for neuron operation in this paper. In supplementary figure 3, even 2.4 V cannot induce the phase transition in anti-ferroelectric layer. The further experiment of phase transition of anti-ferroelectric layer depending on the voltage should be provided.
5. In this paper, authors simulated SNN using the parameter of anti-ferroelectric transistor. However, authors did not provide the parameters which were used for simulation. In addition, authors claimed that threshold level and variation can affect the accuracy of network. However, the reason for the degradation of accuracy was not explained. More information and explanation of SNN simulation based on anti-ferroelectric neurons should be provided.
6. Authors need to cite recent advances in the field of ferroelectric transistors.

Comments from Reviewer 1

Comment 1:

The authors claim to demonstrate for the first time an anti-ferroelectric FET (AFeFET) based neuron. In general the topic is interesting and timely, but in my eyes the manuscript lacks significant novelty.

FeFET based neurons have been proposed already by [44] and others. In these works the main ingredient-accumulative switching has been already demonstrated. However, in the current manuscript under review this aspect stays somewhat vague.

Reply to Comment 1:

Thanks for your insightful advice. Integration and leaky are two fundamental features of artificial neurons. Using the intrinsic physical dynamics to emulate these two features are key merits of emerging devices-based compact neurons. The main innovation of this work is introducing the anti-ferroelectrics' intrinsic physical dynamics to emulate the neuron's features for the first time.

The anti-ferroelectric has no remain polarization, which is the main difference from the ferroelectrics. Correspondingly, the AFeFET is volatile and the FeFET is nonvolatile. As the reviewer mentioned, FeFET based neurons have been proposed as artificial neurons. The neuron based on FeFET lacks the dynamic process of leaky, and need a feedback path¹⁻³ for reset operation. This will significantly increase the hardware cost and energy consumption of neuron implementation. Although, special design of ferroelectric layer (partially crystallization) can give certain leaky property to FeFET^{4,5}, the special designed leaky FeFET neurons needs several seconds time to be operated (integration or leakage), which will increase energy consumption. While the AFeFET is volatile and can achieve spontaneous recovery without any additional reset feedback path. Considering the hardware cost and energy consumption, the AFeFET is more suitable for the mimicry of neurons.

Therefore, we believe that the AFeFET neuron based on the intrinsic integration and leaky properties of anti-ferroelectric is different to previous FeFET neurons in mechanism significantly.

Comment 2:

Fig. 1c shows no remanent polarization at 0 V. However, Fig. 1d shows a remaining MW during sweep. What causes that threshold voltage change - if not the polarization? Also in Fig. 1d it is shown that 1st and 100th cycle result in the very same $I_D V_G$. But in Fig. 1e subsequent pulses show increasing I_d . That data doesn't fit well together. I would encourage authors to add a plot like Fig. 1d but with sweeps just from 0 V to 1.5 V keeping out the neg. voltages to demonstrate the integration behavior.

Reply to Comment 2:

Thanks for the great suggestions and encouragement. The threshold voltage change in Fig. 1d is indeed caused by the polarization of AFeFET, but this change is volatile. Fig. 1c shows the typical double hysteresis of AFE capacitor. As the reviewer mentioned, the macroscopic remanent polarization of AFE capacitor is nearly zero at 0 V, indicating its volatility. This is the feature of AFE which distinguishing from nonvolatile FE. However, the similarity between AFE and FE materials is that they can both be polarized by the electric field, which can be indicated by the window of hysteresis in Fig. 1c. When the applied electric field removed, the polarization of AFE returns to zero. The electric field-induced polarization accounts for the MW of the transfer curve of AFeFET in Fig. 1d. The similar transfer characteristic of AFeFET has been reported by Chengji Jin (*Jin, C. et al., 2018 IEDM, 31.35.31-31.35.34*)⁶.

In Fig. 1d, the 100 transfer curves are achieved under DC sweeping mode and the time interval between cycles (about seconds level) is much longer than that (100 μ s) of pulses in Fig. 1e. In other words, the time interval between cycles under DC sweeping mode is sufficient for self-recovery of AFE. In Fig. 1e, the AFE materials are polarized partially and gradually as the subsequent pulses applied. The time interval between pulses is not sufficient for self-recovery of AFeFET, causing the I_d increase gradually.

The consecutive transfer curves with sweeps from 0 V to 1.5 V keeping out the negative voltages is shown in Fig. R1. Indeed, there is no difference in the transfer curves with or without negative voltages due to the channels of NMOS transistor can only modulate by the polarized charges under the positive gate stimuli.

Figure R1 The transfer curve of AFeFET with sweeps just from 0 V to 1.5 V keeping out the negative voltages.

Comment 3:

The second most important ingredient is the additional functionality of ‘self-reset’ after firing, which however is not sufficiently demonstrated in the manuscript. Authors don’t explain how the neuron would generate a single spike at the output. Moreover, typically the recovery period means a time span where the neuron cannot be excited anymore. But from the data shown it is not obvious how such refractory period can be attained by using only this single device. It seems that authors just stop applying further spikes to the gate after the threshold is reached. It is not shown, how the AFeFET behaves under continuous applied pulses even after reaching a current threshold and sending one spike. Authors should explain what additional circuitry would be mandatory to examine the current threshold, to generate the pulse and to erase the AFeFET.

Reply to Comment 3:

Thank you for the insightful suggestions.

About the generation of spike at the output. Usually, the artificial neuron contains two modules for potential integration and spike generation, respectively. To obtain the driven capability, the spike generation module can be implemented by external comparator⁷⁻⁹. In this work, the AFeFET is used to achieve the potential integration function, and the spike generation module is realized by using Schmidt

trigger as shown in Fig. R2 a. The Schmidt trigger can implement the function of a compactor with less hardware overhead (6 transistors). The threshold voltages (V_{T-} and V_{T+}) of Schmidt trigger are defined by the working parameters of M1, M2, M4 and M5. The R1 resistor is used to convert the current signal of AFeFET into voltage. Once the voltage of V_{int} is higher than V_{T+} , the output potential of Schmidt trigger will be pull up to V_{DD} , as shown in Fig. R2 b. When the voltage of V_{int} is lower than V_{T-} , the output potential of Schmidt trigger will be pull down to ground.

About the controllable refractory period. Actually, most neurons based on emerging devices need peripheral circuits to achieve controllable refractory period¹⁰. In order to demonstrate the refractory period of AFeFET neurons, we have designed peripheral circuits as shown in Fig. R2 a. Once the voltage of V_{int} is higher than V_{T+} , the output potential of Schmidt trigger will be pull up to V_{DD} . The channel of M7 transistor opens when it received the output feedback signal and the input node V_{in} of AFeFET neurons is tied to ground, as shown in Fig. R2 b. Thus, the AFeFET neuron implements the controllable refractory period. The time of refractory period is controlled by the window between V_{T+} and V_{T-} of Schmidt trigger and the leaky time of AFeFET.

To present the complete circuit design of AFeFET neuron, we add Fig. R2 as Supplementary Fig. 11. We also add the description of the spike generation and the realization of the controllable refractory period in the supporting information. In addition, we add the introduction and evaluation of the AFeFET neuron circuit in manuscript as follows:

“The basic integration and fire functionality of the proposed neuron can be achieved by only one AFeFET, while the examination of current threshold, the generation of output spike and controllable refractory period need additional circuits as shown in Supplementary Fig. 11. The detail of this circuit design is described in the supporting information.”

Figure R2 The circuit realization and simulation results of a AFeFET neuron with controllable refractory period and driven capability.

Comment 4:

The system evaluation seems to be based on model assumptions that are not backed by the experiment (see my comments above).

Reply to Comment 4:

Thank you for the comment. The system evaluation based on experiment is the challenge in the field of neuromorphic computing, facing many challenges in large-scale integrated technology and system test hardware. In this work, we focus on the implement of basic neuronal functions, the simulated system evaluation is used to

demonstrate its applicability in system. In the simulation, the data is extracted from the experimental data of AFeFET neuron. In addition, the SNN simulation method has been applied to the system evaluation of emerging neurons in previous works^{11,12}, and the simulation results are sufficient to support its system evaluation.

Comments from Reviewer 2

Overall Comment:

This paper shows a leaky integrate and fire neuron based on antiferro FeFET. Volatility of the device (due to spontaneous depolarization) and spontaneous RESET allows to reproduce neuron leaky behavior, then avoiding the integration of extra capacitances or RESET peripheral circuits. The device is used in two layers SNN for MNIST applications. Low consumption and good endurance is demonstrated.

The paper is complete and presents various topics: material characterization, device electrical characterization and circuit simulation. The neuron emulation is highlighted and well presented. The paper is very clearly written, neuron behavior is clearly described what is helpful for device scientists who may not be expert in the field of brain inspired circuits. The paper is well documented and supplementary materials provide good complementary data. This work would be beneficial for the community as it proposes a device (AFeFET) showing very promising features (volatility, low endurance...) for LIF neuron emulation.

Some points would need clarification before publication. The paper could thus be published provided that major revisions are provided.

Reply to Overall Comment:

We greatly thank the reviewer for the positive comments on this work. We have revised our manuscript carefully according to the reviewer's comments one by one.

Comment 1:

In the abstract, the authors claim 37 fJ/spike, what corresponds to 50 nA threshold (fig.3.g). Then the authors claim 96.8% of recognition rate on MNIST, but this corresponds to I_{th} of 2 μ A (according to fig.4). I think it is misleading as these two features do not correspond to the same operating conditions.

Reply to Comment 1:

Thanks for your insightful advice. We have modified the abstract in our manuscript as follows:

“Moreover, the AFeFET neuron exhibits other comprehensive merits, such as low energy consumption (37 fJ/spike), excellent endurance ($>10^{12}$), high uniformity and high stability. We further construct a two-layer fully ferroelectric (784×400×10) SNN combining established FeFET synapse, the recognition accuracy on *MNIST* datasets is controllable and achieving the maximum (96.8%) under 2 μ A threshold current of AFeFET neuron firing. This work opens the way to emulate spiking neurons with anti-ferroelectric materials and provides a more competitive approach to build highly efficient neuromorphic hardware systems.”

Comment 2:

Volatility

Zr content controls the ferro behavior of HfZrO films, in particular this allows to achieve FE or AFE behaviors. Would it be possible to control the retention time, and thus the neuron characteristics (in particular the speed of leaky behavior) playing with the % of Zr?

Reply to Comment 2:

Thanks for your suggestive comments. Indeed, the Zr content controls the ferro behavior of HfZrO films as shown in Supplementary Fig. 2. HfZrO films exhibit paraelectric-ferroelectric-antiferroelectric evolution with the increasing Zr content. For neurons based on paraelectric (the % of Zr \approx 0%), there is no integration due to it doesn't have polarization. For neurons based on ferroelectric (the % of Zr: 30%-75%), there is no leaky behavior and they can maintain long time even years due to its nonvolatility (Dutta, S. et al., 2019 Symposium on VLSI Technology, T140-T141)². For neurons based on anti-ferroelectric (the % of Zr: 75%-100%), the leaky behavior is inherent characteristic due to its volatility. Thus, we think it is possible to control the retention time by playing with the % of Zr, but the detailed mechanisms need further study. We chose 80% Zr content considering the completely leaky property and the relatively large window of electric field-induced polarization.

Comment 3:

RESET

How is the neuron RESET after firing? What is the corresponding required overhead in the circuit?

Reply to Comment 3:

Thank you for the insightful suggestion. The neuron resets without additional reset circuits due to the volatility of AFeFET, but needs a circuit to supply the refractory period. Fig. R3 shows the entire neuron circuits, including integration and spike generation modules. The potential integration function is achieved by the AFeFET and the spike generation module is realized by Schmidt trigger. The threshold voltages (V_{T-} and V_{T+}) of Schmidt trigger are defined by the working parameters of M1, M2, M4 and M5 transistor. The R1 resistor is used to convert the current signal of AFeFET into voltage. Once the voltage of V_{int} is higher than V_{T+} , the output potential of Schmidt trigger will be pull up to V_{DD} . When the voltage of V_{int} is lower than V_{T-} , the output potential of Schmidt trigger will be pull down to ground. Thus, the output spike is generated. The channel of M7 transistor opens when it received the output feedback signal and the input node V_{in} of AFeFET neuron is tied to ground. Thus, the AFeFET neuron implements the controllable refractory period. The time of refractory period is controlled by the window between V_{T+} and V_{T-} of Schmidt trigger and the leaky time of AFeFET.

Figure R3 The circuit realization of AFeFET neuron with controllable refractory period and driven capability.

Comment 4:

Circuit size

AFeFET are three terminal devices. They are thus bigger than other emerging devices using 2 terminals. I do not think it is a big issue as number of neurons is less important than number of synapses. However, the authors also use a 3 terminal (FeFET) for synapse emulation. Could the authors comment on the impact it can have on the total circuit size?

Reply to Comment 4:

Thanks for the comment. Yes, AFeFET is larger than the 2-terminals emerging devices. However, AFeFET based neurons do not need additional capacitors and reset feedback circuit, which will lead to smaller neuron circuit size. In this work, the AFeFET based neurons would fit most of the synapses which based on emerging devices, including three terminal devices and two terminal devices. We chose FeFET for synapse emulation from the material technology perspective. The FeFET synapses and AFeFET neurons are in the same material system (doped HfO₂), which is conducive to the implementation of chip.

Comment 5:

Current saturation

In page 8, the authors propose interpretation to the saturation of I_d during gate stimuli. I am confused by the interpretation. Indeed, the $I_d(t)$ characteristics looks like typical $I_d(V_g)$ MOSFET curve. As pulses are applied, electric field in the Fe layer increases, acting as a control gate. Thus, $I_d(\text{time})$ curve of fig.2.i could be transferred as a $I_d(V_g)$ curve. Thus, the saturation could be explained by the $I_d(V_g)$ saturation behavior that is measured at high V_g , and I do not see how it is possible to decorrelate the saturation from the transistor channel conduction to the saturation of domain switching. One option may be to test a reference device with same equivalent oxide thickness for the gate oxide but having non ferro behavior. I think it is important to clarify this point to distinguish between current vs ferroelectric saturation.

Reply to Comment 5:

Thanks for your insightful comments. As the reviewer pointed, the $I_d(t)$ characteristics looks like the typical $I_d(V_g)$ MOSFET curve. But, there are fundamental differences between them, as shown in Fig. 2.i and Supplementary Fig. 3.c. For clarity, we put them in one figure as shown in Fig. R4 a-b. The typical $I_d(V_g)$ MOSFET curve is the tendencies of I_d under increasing V_g amplitude, and the I_d saturated under high V_g amplitude. However, $I_d(t)$ curve is the tendencies of I_d under gate pulses with same V_g amplitude. With a series of same amplitude V_g pulses, the AFE layer is polarized and the polarization charges control the transistor channel. The positive polarization charges accumulate on the side AFE layer, inducing the accumulation of negative charges in the transistor channel and thus lead to the increasement of I_d . With the sustainable growth of gate pulse amounts, the AFE layer tends to saturate and polarization charges stop to increase. As a result, the I_d stop to increase.

In addition, the function of AFE control layer can be demonstrated by the memory window of the transfer curves of AF₂FET in Fig. R4 c, which show a typical hysteresis curve different from Fig. R4 a. In order to distinguish between current vs ferroelectric saturation, we measured a reference MOSFET device and 22 pF capacitor (the capacitance is equivalent to that of AFE layer) as shown in Fig. R4 d. The $I_d(t)$ has not integration and saturation behavior.

Figure R4 **a** The typical transfer curves of MOSFET. **b** The $I_d(t)$ curve of AFeFET under continuous gate pulses with different amplitudes (1.3 V-1.8 V amplitude, fixed 100 μ s interval, fixed 100 μ s width). **c** The transfer curves of AFeFET. **d** The $I_d(t)$ curve of MOSFET with 22 pF capacitor.

Comment 6:

Current threshold for firing

Various current threshold values are presented in the article. Fig.1 uses 1 μ A and fig.4 shows the threshold values affects recognition rate. Looking at Fig.2, it seems that $I_{th} \sim \mu$ A is in the sub-threshold regime of the $I_d(V_g)$ characteristics of the transistor. I was this wondering if detecting the threshold in the over threshold regime of the transistor could help in reducing potential variability issue on the number of required pulses to reach a certain current level. As a consequence, I think it would be interesting to quantify the impact of I_{th} value (in sub vs over threshold regime) on the device to device variability of the firing event (ie variability of pulse number for firing vs I_{th} value).

Reply to Comment 6:

Thanks for your suggestion. Figure R5 shows the variability of pulse number for firing under 1 μ A and 5 μ A threshold, respectively. The statistical data were extracted from 50 cycles for every device. The pulses with 1.7 V amplitude, 100 μ s interval and

100 μ s width are applied. From the results, we find that the variability will increase with increasing I_{th} value. As the pulses are applied continuously, the output current increased, but the current increment under one pulse decreases. This is due to almost of the AFE domains, which can be reversible under 1.7 V pulse, have been reversed by the earlier pulses. If detecting the I_{th} in the over threshold regime, the current increment is tiny and the current tends to saturation, thus the variation of required pulse number for reaching I_{th} increases. In addition, the higher I_{th} means higher energy consumption, this is not we expected.

Figure R5 Statistical data of input pulse numbers of AFeFET neuron firing for different threshold.

Comment 7:

Network hardware implementation (page 12)

In the hardware implementation of the network, during inference, input signal is applied to the gates of the synapses. What signal is applied to the BL? How different would it be to send input signals to the BL?

Reply to Comment 7:

Thank you for this comment. We are sorry for the mistake in the manuscript. Indeed, the modulated signal is applied to the gates of FeFET synapses (WLs) during training. During inference, the input signal is applied to the drains of FeFET synapses (BLs).

The corresponding text in the manuscript has been revised as follows: “During inference, the input signal is applied to the drains of FeFET synapses (BLs), pulse-width modulators (PWMs) collect current on source lines (SLs) and convert to pulses with fixed amplitude and various widths.”

We also updated the Fig. 4(a) to show the schematic of hardware implementation of the network as shown in Fig. R6.

Figure R6 The schematic of two-layer fully ferroelectric SNN (784×400×10) for classifying *MNIST* datasets and the proposed hardware implementation of the network based on FeFET synapses and AFeFET neurons.

Comment 8:

Recognition rate

Networks aiming at character recognition from MNIST data sets are known to be quite resilient to device variability. Could the authors comment on what would happen if the number of neurons in the hidden layer is decreased, would the system be more sensitive to device imperfections?

Reply to Comment 8:

We thank the referee for this comment. To study the influence of hidden layer neurons on the network performance, we performed a simulation on the network with $784 \times 200 \times 10$ and $784 \times 100 \times 10$, respectively. Fig. R7 a shows the inference accuracy as a function of training epochs under different network structures. The results show that when the hidden neuron numbers decrease, the training speed and the final accuracy decrease. Compared to the network with $784 \times 400 \times 10$ configuration, the network performance decreased by about 2% when the hidden neuron number

decreased by half. But the accuracy results are still acceptable under those conditions that do not pursue high accuracy but need lower hardware overhead. We also studied the network degradation under different threshold variations during inference, as shown in Fig. R7 b. The results illustrate that, similar to that with 400 hidden neurons, the network has a certain resilience to device variability. The network with fewer hidden neurons shows a faster deterioration speed, just as the reviewer comments that the system with fewer hidden neuron numbers is more sensitive to device's imperfections.

Figure R7 a Inference accuracy as a function of training epochs under different network structures. **b** The robustness of network with threshold variations.

Comment 9:

Reference

Page 3, line 3: references should be added on structures combining emerging memories with capacitors for neuron implementation.

Reply to Comment 9:

Thank you for the insightful suggestions. We have added references in page 3, which are neuronal researches based on emerging memories with capacitors (*Wu, Z. et al., 2020, Adv. Mater. 32, 2004398. Wang, Z. et al., 2018, Nat. Electron. 1, 137-145. Yi, W. et al., 2018, Nat. Commun. 9, 4661.*).

Comments from Reviewer 3

Overall Comment:

In this paper, authors demonstrated a leaky integrate-and-fire (LIF) neuron based on an anti-ferroelectric transistor. The device-level performance was systematically demonstrated. In addition, authors simulated the SNN to demonstrate the feasibility of anti-ferroelectric transistor neurons. However, in this manuscript, the issues of anti-ferroelectric materials for devices were not discussed. In my view, this work is interesting but can be further improved through these comments.

Reply to Overall Comment:

We greatly thank the reviewer for the positive comments on this work. We have revised the manuscript carefully according to the reviewer's comments one by one.

Comment 1:

In this paper, the phase transition of $\text{Hf}_{0.2}\text{Zr}_{0.8}\text{O}_x$ layer by electric field was used to emulate the LIF characteristics. $\text{Hf}_{0.2}\text{Zr}_{0.8}\text{O}_x$ layer seems to have the potential as neuron devices because of the similarity between intrinsic polarization/depolarization of $\text{Hf}_{0.2}\text{Zr}_{0.8}\text{O}_x$ layer and the dynamic process of membrane in biological neuron. However, $\text{Hf}_{0.2}\text{Zr}_{0.8}\text{O}_x$ layer can show the permanent phase transition from t-phase to o-phase during repeated operation (*Adv. Electron. Mater.* 6, 2000631, (2020), *ACS Appl. Mater. Interfaces* 8, 15466-15475 (2016)). This issue should be discussed.

Reply to Comment 1:

Thank you for the great advices. These two works are reached by the team of Prof. C.S. Hwang (*Adv. Electron. Mater.* 6, 2000631, (2020), *ACS Appl. Mater. Interfaces* 8, 15466-15475 (2016))^{13,14}. In these two works, researchers have investigated different Zr contents ($x=0.26-0.70$) in $\text{Hf}_{1-x}\text{Zr}_x\text{O}$ film. Thereported results indicated that t-phase transforms to o-phase during repeated operation. In the pristine state, the ration of t-phase and the double variable polarization ($2P_v$) of $\text{Hf}_{1-x}\text{Zr}_x\text{O}$ increased as the increasing Zr content. The ration of t-phase ($2P_v$) in $\text{Hf}_{0.49}\text{Zr}_{0.51}\text{O}_2$, $\text{Hf}_{0.40}\text{Zr}_{0.60}\text{O}_2$, and $\text{Hf}_{0.30}\text{Zr}_{0.70}\text{O}_2$ is 20%, 65%, 95%, respectively. After 10^6 cycles, the ration of t-phase ($2P_v$) also increased with the increasing Zr content, and the increment increased gradually. The ration in $\text{Hf}_{0.49}\text{Zr}_{0.51}\text{O}_2$, $\text{Hf}_{0.40}\text{Zr}_{0.60}\text{O}_2$, and $\text{Hf}_{0.30}\text{Zr}_{0.70}\text{O}_2$ is 0.5%, 25% 81%, respectively after 10^6 cycles. However, we can find that the permanent t-o phase transition weakened gradually with the increasement of Zr content ($\text{Hf}_{0.49}\text{Zr}_{0.51}\text{O}_2$:

97.5%, Hf_{0.40}Zr_{0.60}O₂: 61%, Hf_{0.30}Zr_{0.70}O₂: 14.7%). We believe that the permanent t-o phase transition will be slightly with the Zr content further increased. In our work, we get the same tendency of Hf_{1-x}Zr_xO films with various Zr contents, but the range of Zr contents is larger than that in these two researches. In our work, the AFeFET is based on Hf_{0.2}Zr_{0.8}O₂, whose t-phase ration will be higher and the permanent t-o phase transition will be indistinctive than Hf_{0.30}Zr_{0.70}O₂ (14.7%) after repeat operations.

In addition, the AFeFET in this work is working in subthreshold region due to its operating electric field (1.5-1.7 MV/cm) is much smaller than that in the referred two researches (3-3.26 MV/cm). The small electric field cannot operate AFE films to saturation and the possibility of permanent t-o phase transition will be smaller.

According to the aforementioned points, we believe that the permanent t-o phase transition is negligible in Hf_{0.2}Zr_{0.8}O₂ AFeFET under relative low operating voltage.

Comment 2:

As mentioned in comment 1, the permanent phase transistor from t-phase to o-phase in Hf_{0.2}Zr_{0.8}O_x layer can be induced during repeated operation. This phenomenon can cause the cycle-to-cycle variation issue. Authors claimed that anti-ferroelectric layer did not show the change during 10¹² cycles in Fig. 3h. Is there any reason for the absence of the phase transition from t-phase to o-phase during repeated operation? In addition, anti-ferroelectric neurons showed the degradation of output current level after only 5×10⁵ cycles. It seems that anti-ferroelectric neurons cannot be fired afterwards.

Reply to Comment 2:

Thank you for this comment. We measured the endurance of AFE MIM structure which is the endurance bottleneck of AFeFET in order to speed up the measurement. we can find that there is slight degradation which can be observed by the polarization @ 0 V after 10¹⁰ cycles in Fig. 3h. This demonstrates that there is slight permanent phase t-o transition after 10¹⁰ cycles. But, 3 MV/cm was used in the endurance measurement of AFE capacitor in order to observe the hysteresis clearly. The large electric field will operate AFE films to saturation and the permanent t-o phase transition tends to occur, thus the polarization @ 0 V show slight degradation.

Whereas, the applied operating electric field is only 1.5-1.7 MV/cm in the AFeFET structure, we find that the output currents show variation other than degradation. In order to evaluate the variation of output current, the statistical output currents vs cycles (10^5) are shown in Fig. R8. The 12 gate pulses with 1.5 V amplitude, 100 μ s interval and 100 μ s width were applied on the AFeFET. From the statistical results, the output currents present fluctuation other than degradation. The fluctuation of output currents accounts for the variation of the required pulse number in Fig. 3d-f. To avoid confusing the reader, we modify the supplementary Fig. 10a as shown in Fig. R9.

Figure R8 Statistical data of the output currents of AFeFET neurons with 12 gate pulses (1.5 V amplitude, 100 μ s interval and 100 μ s width).

Figure R9 The AFeFET neuron can fire 5×10^5 cycles stably without any significant deterioration.

Comment 3:

Authors used NMOS transistor and anti-ferroelectric capacitor to implement anti-ferroelectric transistor. Anti-ferroelectric capacitor was formed on the gate pad of NMOS transistor. In this case, the applied voltage can be divided into the gate dielectric layer of NMOS transistor and anti-ferroelectric layer depending on the capacitance of layers. However, the information of NMOS transistor was missing. The information of NMOS transistor, such as thickness of dielectric and area of gate stack, should be added. In addition, in Fig. 1d and 2h, schematic diagrams of device structure were shown as if anti-ferroelectric capacitor was directly stacked on NMOS transistor.

Reply to Comment 3:

Thank you for the insightful suggestions. The W/L of NMOS is $10\mu\text{m}/1\mu\text{m}$. The dielectric of NMOS is 4 nm. We add these information of NMOS in sample fabrication as follows: “The W/L of NMOS is $10\mu\text{m}/1\mu\text{m}$ and its dielectric thickness is 4 nm.” In addition, we modify the schematic of AFeFET in Fig. 1d and 2h.

Comment 4:

As mentioned in comment 3, the applied gate voltage will be divided into the gate dielectric layer of NMOS and anti-ferroelectric layer. Therefore, the applied voltage to anti-ferroelectric layer seems to be lower than 1.5 V, which is average voltage for neuron operation in this paper. In supplementary Fig. 3, even 2.4 V cannot induce the phase transition in anti-ferroelectric layer. The further experiment of phase transition of anti-ferroelectric layer depending on the voltage should be provided.

Reply to Comment 4:

Thank you for the insightful suggestions. As mentioned by reviewer, the applied voltage can be divided into the gate dielectric layer of NMOS transistor and anti-ferroelectric layer. The capacitance and resistance of anti-ferroelectric layer (C_{AFE}) are about 10-17 pF and 10 G Ω , respectively. While, capacitance and resistance of NMOS (C_{gs}) are about 1 pF and 200 M Ω respectively. In addition, there is about inherent 15 pF parasitic capacitance in this BEOL structure AFeFET. Based on these parameters,

we have built a equivalent complex impedance model and calculate that about 1.0 V is applied on the AFE layer. In addition, the $I_d(t)$ in Fig.2i also demonstrates that most voltage is applied on the AFE layer due to there is no I_d under the first few pulses.

In supplementary Fig. 3b, the hysteresis loop of AFE capacitor was obtained by the I-V curves under triangle pulses with 200 μs period as shown in Fig. R10 a. Large voltage is required to reverse all domains under single pulse. But the existence of polarization current peak (the red arrow) indicates that partial domains can be reversed even under 1.4 V pulse in Fig. R10 b. To observe the polarization current peak clearly, rectangular pulses with shorter time period (1 μs width, 300 ns rising and falling edges) were applied to AFE capacitor as shown in Fig. R10 c. The polarization current peak of the falling edge (the red arrow) indicates the AFE capacitor can induce the phase transition even under 0.9 V operating voltage.

Figure R10 The I-V curves of AFeFET capacitor **a** under triangle pulse with 1.4 V-2.6 V amplitudes. **b** under triangle pulse with 1.4 V amplitude. The existence of polarization current peak indicates that the device can work under 1.4 V pulse. **c** under rectangular pulses with shorter time period (1 μs width, 300 ns rising and

falling edges).

Comment 5:

In this paper, authors simulated SNN using the parameter of anti-ferroelectric transistor. However, authors did not provide the parameters which were used for simulation. In addition, authors claimed that threshold level and variation can affect the accuracy of network. However, the reason for the degradation of accuracy was not explained. More information and explanation of SNN simulation based on anti-ferroelectric neurons should be provided.

Reply to Comment 5:

In the proposed SNN, key device parameters include the pulse number for firing, the cycle to cycle variation, and the leaky time constant have been added. In this work, the pulse number (64) for firing under 10 μs pulse width is used due to the highest number counts that correspond to the bits number (6 bits) of the neuron's membrane potentials. The 9% cycle to cycle variation is extracted from the statistical data of 10 μs pulse width in Fig. 3e. In addition, the leaky time constant (800 μs) is extracted from the integrate-and-fire process under 10 μs pulse width (Supplementary Fig. 7b). Based on these three parameters, we performed our simulations. To clearly present this, we polish the description of the *SNN simulation method* in the *Methods* as follows:

“The pulse number (64) for firing under 10 μs pulse width is used due to the highest number counts that correspond to the bits number (6 bits) of the neuron's membrane potentials. The 9% cycle to cycle variation is extracted from the statistical data of 10 μs pulse width in Fig. 3e. In addition, the leaky time constant (800 μs) is extracted from the integrate-and fire process under 10 μs pulse width (Supplementary Fig. 7b). The energy consumption was calculated according to the data of 10 μs pulse width in Fig. 3g.”

Just as we claimed, the pulse number for firing is used for simulation, which is equivalent to the number of the membrane potential. When decreasing the threshold, the number of the potential membranes decreases, corresponding to the degradation of the precision of the membrane potential. Thus, with decreasing the threshold, the

recognition accuracy decreases. In addition, after training, the inference accuracy only decreases 1% even the variation increases to $\pm 16.7\%$, as we claimed in the manuscript. This is because a 5% variation is considered during the training process. When the variations further increase, the accuracy shows obvious degradation. This is because a higher variation makes the threshold value far from the training value, inducing the firing time to deviate from the ideal firing time, then the accuracy decreases.

To clearly present this, we modified our manuscript in page 12 as follows: “The pulse number for firing is equivalent to the number of the membrane potential during training. When increasing the threshold, the number of the potential membranes increases, corresponding to the increasing precision of the membrane potential. Thus, with increasing the threshold, the recognition accuracy increases.”

Comment 6:

Authors need to cite recent advances in the field of ferroelectric transistors.

Reply to Comment 6:

Thanks for the suggestion. We add recent advances of ferroelectric field-effect transistors as reference 33 and reference 60. (*Mulaosmanovic, H. et al., 2021, Nanotechnology, 32(50). Sun, C. et al., in 2021 IEEE Symposium on VLSI Technology, 2021*).

References:

1. Wang, Z. et al. Experimental demonstration of ferroelectric spiking neurons for unsupervised clustering. in *2018 IEEE International Electron Devices Meeting (IEDM)* 13.13.11-13.13.14, (2018).
2. Dutta, S. et al. Biologically plausible ferroelectric quasi-leaky integrate and fire neuron. in *2019 Symposium on VLSI Technology* T140-T141, (2019).
3. Dutta, S. et al. Supervised learning in all FeFET-based spiking neural network: opportunities and challenges. *Front. Neurosci.* **14**, 634,(2020).
4. Chen, C. et al. Bio-inspired neurons based on novel leaky-FeFET with ultra-low hardware cost and advanced functionality for all-ferroelectric neural network. in *2019 Symposium on VLSI Technology* T136-T137, (2019).
5. Luo, J. et al. Capacitor-less stochastic leaky-FeFET neuron of both excitatory and inhibitory connections for SNN with reduced hardware cost. in *2019 IEEE International Electron Devices Meeting (IEDM)* 6.4.1-6.4.4, (2019).

6. Jin, C. et al. Experimental study on the role of polarization switching in subthreshold characteristics of HfO₂-based ferroelectric and anti-ferroelectric FET. in *2018 IEEE International Electron Devices Meeting (IEDM)* 31.35.31-31.35.34, (2018).
7. Munoz-Martin, I. et al. Hardware implementation of PCM-based neurons with self-regulating threshold for homeostatic scaling in unsupervised learning. in *2020 IEEE International Symposium on Circuits and Systems (ISCAS)* 1-5, (2020).
8. Huang, H. M. et al. Quasi-Hodgkin-Huxley neurons with leaky integrate-and-fire Functions physically realized with memristive devices. *Adv. Mater.* **31**, e1803849, (2019).
9. Wright, C. D. et al. Beyond von-neumann computing with nanoscale phase-change memory devices. *Adv. Funct. Mater.* **23**, 2248-2254, (2012).
10. Zhang, X. et al. An artificial neuron based on a threshold switching memristor. *IEEE Electron Device Lett.* **39**, 308-311, (2018).
11. Wang, W. et al. A self-rectification and quasi-linear analogue memristor for artificial neural networks. *IEEE Electron Device Lett.* **40**, 1407-1410, (2019).
12. Zhang, X. et al. Experimental demonstration of conversion-based SNNs with 1T1R Mott neurons for neuromorphic Inference. in *2019 IEEE International Electron Devices Meeting (IEDM)* 6.7.1-6.7.4, (2019).
13. Hyun, S. D. et al. Field-Induced ferroelectric Hf_{1-x}Zr_xO₂ thin films for high-k dynamic random access memory. *Adv. Electron. Mater.* **6**, 2000631, (2020).
14. Park, M. H. et al. Effect of Zr content on the wake-up effect in Hf_{1-x}Zr_xO₂ films. *ACS Appl. Mater. Interfaces* **8**, 15466-15475, (2016).

REVIEWER COMMENTS

Reviewer #1 (Remarks to the Author):

I thank the authors for the responses to my questions. I agree that the AFeFET concept differs from the FeFET-based neurons and it would be a benefit discussing this work and the additional aspects such as self-reset possibility in the AFeFET within the community. Unfortunately, I am still not really convinced about the dynamic functionality of the AFeFET and think that further explanation or data would be mandatory to add to the manuscript before publication as detailed below.

Still there is a gap between the DC-characteristics and the dynamic characteristics of the device. In their response to my comment 2 from the first review authors explain the differences between DC-behavior and transient behavior by the very different timescales of the measurements. However, from the data given there is no evidence that the integration behavior can be really attributed to the accumulative switching of AFE phase into a ferroelectric phase. E.g. the depicted behavior of the device in Fig. 1e could be also explained by a charging of the internal node forming between the transistors gate and the AFE-capacitor via leakage through the AFE capacitor. I am far from questioning in general the concept of this AFeFET and potentially use in a neuron circuit. However, in order to prove the accumulative dynamics, transient behaviour of the AFE-layer itself should be analyzed. E.g. transient measurements of the AFE-capacitors polarization behavior would be mandatory to rule out any charging effects. The same holds true for the refractory period in my next comment.

In the revised manuscript authors explain now more clearly that the refractory period is induced by the additional circuit but cannot necessarily be attributed to the AFeFETs dynamic behavior itself. The new supplementary figure 11 also explains now very well the approach of implementing the refractory period. However, it also becomes clear that this is realized by simply shorting the input to gnd, which for the AFeFET is the same effect as just pausing the input pulses, and in a real circuit would actually largely increase the power consumption (input short to gnd). However, evidence of the origin of the leaky-behavior of the AFeFET and its physical origin e.g. by a relaxation of the ferroelectric phase back to a AFE phase is not really shown. The decay of V_{int} in Fig. 11 during the refractory period could be again explained by either a discharge of this node via R1 while the AFeFET is switched off by shorting the gate of the AFeFET to gnd, or by the discharging effect of the internal floating gate node between the transistor and the AFE capacitor. Such effects should be ruled out. Hence again, transient measurements of the AFE-capacitors polarization behavior also showing the polarization decay would be mandatory.

Finally, now in view of the given circuit in the new supplementary figure 11 the value of R1 and estimation of the power consumption is unclear to me. Authors claim in sup. figure 9 a power consumption of 37 fJ per spike at a 50nA threshold for I_D . But firstly, from the I_D curves given in this figure such low threshold seems not realistic (close to noise level). Secondly, what would be the value of

R1? It has to be large enough to enable voltage drop at low ID but has to be low enough to guarantee discharge of V_{int} when the AFeFET is switched off. Thus the question arises: what is the influence of R1 on the whole dynamics of the Neuron?

In summary, I still would suggest to solve this mentioned issues before publishing the work.

Reviewer #3 (Remarks to the Author):

Authors well revised their manuscript according to reviewers' comments. In my opinion now it is acceptable for publication in this journal.

I. Comments from Reviewer 1

Comment 1:

Still there is a gap between the DC-characteristics and the dynamic characteristics of the device. In their response to my comment 2 from the first review authors explain the differences between DC-behavior and transient behavior by the very different timescales of the measurements. However, from the data given there is no evidence that the integration behavior can be really attributed to the accumulative switching of AFE phase into a ferroelectric phase. E.g. the depicted behavior of the device in Fig. 1e could be also explained by a charging of the internal node forming between the transistors gate and the AFE-capacitor via leakage through the AFE capacitor. I am far from questioning in general the concept of this AFeFET and potentially use in a neuron circuit. However, in order to prove the accumulative dynamics, transient behaviour of the AFE-layer itself should be analyzed. E.g. transient measurements of the AFE-capacitors polarization behavior would be mandatory to rule out any charging effects. The same holds true for the refractory period in my next comment.

Reply to Comment 1:

Thanks for the insightful advice. Fig. R1 a shows the AFE-layer has integration behavior. The single rectangular pulse with different pulse widths (10 μ s and 10 ms) were applied to AFE capacitor as shown in Fig. R1 a. The polarization current peaks of the AFE capacitor were obtained during the 10 μ s falling edge of rectangular pulses. We can find that the polarization current increases with increasing the pulse width. The polarization current of 10 ms pulse is higher than that of 10 μ s obviously. The results show that the AFE layer can be polarized by applying the 1.5 V pulse for a long time, indicating that the polarization is the possible cause of the integration in Fig. 1 d and e in manuscript.

In order to obtain the transient behavior of the AFE-layer, we have followed Ref. 28 (Dutta, S. et al. *Biologically plausible ferroelectric quasi-leaky integrate and fire neuron. in 2019 Symposium on VLSI Technology T140-T141, (2019)*). A series of pulses

are applied to the M-AFE-M structure and the current showing a slight increasing trend, containing the contribution of the antiferroelectric switching and dielectric response. We deduct the charge caused by the linear capacitor charging effects from total charge, which differs from the Ref. 28 due to antiferroelectric is volatile. From Fig. R1 b, we can find that the nonlinear increase and decrease of total charge is contributed by antiferroelectric switching. The linear changes are caused by the linear capacitor charging effects as shown by the green dotted lines in Fig. R1 b. We get the capacitance value of the charging and discharge parts are 14.3 pF and 16.1 pF from the linear parts, which are in keep with that of measurements.

Then, we get the change in the amount of charge caused by linear capacitor charge and discharge effect (the orange dotted lines). Finally, the polarization of antiferroelectric can be obtained by deducting the charge and discharge effect portion from the total charges. In order to observe the integration behavior of AFE-layer, a series of pulses with 1 μ s width, 20 ns delay and different amplitudes are applied to M-AFE-M devices. Fig. R1 c shows the total current of M-AFE-M devices under ten 2.0 V pulses. We get the total charge by accumulating the total current over time, as shown in Fig. R1 d. Then, we obtain the polarization of antiferroelectric by deducting the charges of charging and discharging effects (Fig. R1 d). The polarization of antiferroelectric exhibits integration behavior under continuous pulses and tends to saturation. Moreover, the polarization recovers during hundreds of nanoseconds which benefited from the volatile of antiferroelectric. Fig. R1 e show the polarization of antiferroelectric under pulses with different amplitudes. The polarization under lager pulses amplitudes integrates faster and tend to saturation earlier, this phenomenon is consistent with that of AF₂FET. Overall, the integration behavior through the AFE polarization switching is clearly observed.

Fig. R1. **a** The integration behavior of M-AFE-M device under rectangular pulses with 10 μs and 10 ms pulse widths, respectively. **b** The current, total charge, the charge trends of charging and discharging effects, **c** The current under a series of pulses, **d** The trends of total charge and polarization, **e** The polarization under different pulse amplitudes of M-AFE-M devices.

Comment 2:

In the revised manuscript authors explain now more clearly that the refractory period is induced by the additional circuit but cannot necessarily be attributed to the AFeFETs dynamic behavior itself. The new supplementary figure 11 also explains now very well the approach of implementing the refractory period. However, it also becomes clear that this is realized by simply shorting the input to gnd, which for the AFeFET is the same effect as just pausing the input pulses, and in a real circuit would actually largely increase the power consumption (input short to gnd). However, evidence of the origin of the leaky-behavior of the AFeFET and its physical origin e.g. by a relaxation of the ferroelectric phase back to a AFE phase is not really shown. The decay of V_{int} in Fig. 11 during the refractory period could be again explained by either a discharge of this node via R1 while the AFeFET is switched off by shorting the gate of the AFeFET

to gnd, or by the discharging effect of the internal floating gate node between the transistor and the AFE capacitor. Such effects should be ruled out. Hence again, transient measurements of the AFE-capacitors polarization behavior also showing the polarization decay would be mandatory.

Reply to Comment 2:

Thanks for the great comment. As reviewer commented, shorting the input to ground does increase the power consumption. In order to solve this problem, we have improved the circuit of AFeFET neuron to implement refractory period, as shown in Fig. R2 a. In the improved circuit, the M7 NMOS in parallel is replaced by PMOS in series. Once the voltage of V_{int} is higher than V_{T+} , the output potential of Schmidt trigger will be pull up to V_{DD} . The channel of M7 PMOS transistor turns off when received the output feedback signal and the AFeFET neuron implements the controllable refractory period. Thus, the power consumption can be decreased during the refractory period. **To decrease the power consumption of circuit, we replace the supplementary Fig. 11 with Fig. R2.**

Fig. R2. The circuit realization and simulation results of a AFeFET neuron with refractory period and driven capability.

In the proposed AFeFET neuron, the drain current represents the integration of membrane potential. The role of $R1$ is to convert the drain current into voltage. Usually, a $5\text{ M}\Omega$ resistor was used and the results was shown in Fig. R2. We think the discharge of V_{int} via $R1$ is not a serious problem in the AFeFET neuron circuit. The discharge process of V_{int} is determined by $R1$ and parasitic capacitance (between source and drain) in the corresponding leakage path, which is on the order of tens of femto-farads. Thus, the leakage time constant is the product of $R1$ resistance and parasitic capacitance, which is tens of nano-seconds typically. So, the tens of nano-seconds leakage time via $R1$ is negligible in the refractory period. But, the discharging effect between the transistor and the AFE capacitor is the major consideration for refractory period. The decay time of AFE capacitor is on the order of hundreds of nano-seconds (Fig. R1),

which is fast than that of AFeFET neuron (ms). Obviously, the discharging effect dominates the refractory period. Overall, the polarization decay of AFE layer determines the leaky capacity of AFeFET, but the leaky speed depends on the discharging effect.

Comment 3:

Finally, now in view of the given circuit in the new supplementary figure 11 the value of R1 and estimation of the power consumption is unclear to me. Authors claim in sup. figure 9 a power consumption of 37 fJ per spike at a 50 nA threshold for I_D . But firstly, from the I_d curves given in this figure such low threshold seems not realistic (close to noise level). Secondly, what would be the value of R1? It has to be large enough to enable voltage drop at low I_D but has to be low enough to guarantee discharge of V_{int} when the AFeFET is switched off. Thus the question arises: what is the influence of R1 on the whole dynamics of the Neuron?

Reply to Comment 3:

Thanks for your comment. The threshold of 50 nA is very close to the measurement noise, but it can still be distinguished by Keysight 1500A measurement system, as shown in Fig. R3. In fact, 50 nA is almost the current measurement limit of this instrument in narrow pulse measurement mode. In this paper, the purpose is to investigate and demonstrate the energy consumption potential of the AFeFET neuron. The results demonstrate the AFeFET has great potential in low-power applications with the development of measurement technology in the future.

Fig. R3. The energy consumption per spike under pulses with 1 μs width and interval.

As mentioned in comment 2, the role of R1 is to convert the drain current into voltage. As the reviewer commented, the value of R1 is important, we have carefully adjusted the value of R1. The value of R1 have influence on the accumulation process of V_{int} and the number of refractory periods, as shown in Fig. R4. Under the same excitation pulse conditions, the faster potential accumulation speed of V_{int} and the higher frequency of refractory period are observed with larger R1. But, we can find that the discharge time of V_{int} is not affected by the R1 because it depends on the leaky of AFeFET.

Fig. R4. The accumulation process of V_{int} with different values of R1.

REVIEWERS' COMMENTS

Reviewer #1 (Remarks to the Author):

Thank you for your answers to my questions and the additional information. From my perspective this gives a reasonable explanation. I don't have further comments.